# HSGG: Training-Free Hierarchical Scene Graph Generation with Geometry-Guided Relation Reasoning

**Yunzhe Liu** [1] **Wenbiao Liu** [1] **Lihui Cen** [2] **Zhe Qu** [1] **Yigang Cen** [3] **Yixiong Liang** [1] **Yongfang Xie** [2] **Shichao Kan*** [1]

## Abstract

Scene Graph Generation (SGG) connects visual perception with structured reasoning, but is limited by scarce annotations and the long-tailed distribution of relational predicates. Training-free methods based on vision-language models (VLMs) reduce supervision requirements, yet often rely on flat graph structures and produce hallucinated relations. We present HSGG, a training-free framework for open-world Hierarchical Scene Graph Generation, whose inference proceeds in two steps. First, bidirectional hierarchical entity perception combines top-down object expansion with bottom-up attribute reasoning to construct multi-level scene hierarchies capturing part–whole semantics. Second, geometry-guided relation reasoning infers valid relations from these structured entities: geometry-aware relation filtering first prunes spatially implausible object pairs using 2D proximity, depth cues, and object scale, and geometry-grounded contrastive relation decoding then suppresses hallucinated predicates by contrasting predictions from a visually grounded expert against a hallucination-prone geometric prior, ensuring relations are both geometrically consistent and semantically coherent. Experiments show that HSGG generalizes effectively to unseen objects and predicates without training, substantially reduces relational hallucinations, and consistently improves downstream reasoning performance.

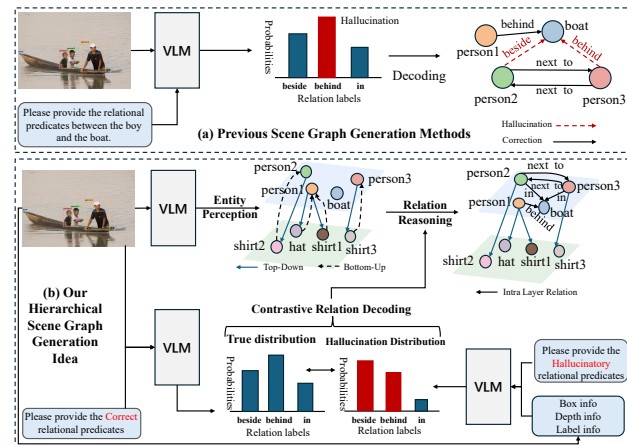

*Figure 1.* Motivation and overview of HSGG. (a) Flat relation decoding in VLM-based SGG leads to hallucinations. (b) HSGG mitigates hallucinations via hierarchical entity perception and geometry-guided contrastive decoding.

## 1. Introduction

Scene Graph Generation (SGG) serves as a crucial bridge between low-level visual perception and high-level structured reasoning. By representing images as graphs of object nodes and their pairwise semantic relations, SGG has demonstrated utility in downstream tasks (Chang et al., 2021; Gothoskar et al., 2021) such as image captioning (Zhang et al., 2025; Lu et al., 2025), visual question answering (Lei et al., 2023), and embodied intelligence (Fan et al., 2025). However, conventional SGG methods rely heavily on dense annotations and struggle with the long-tailed distribution of relational predicates, limiting their generalization to open-world scenarios.

Training-free approaches based on vision-language models (VLMs) have recently emerged to mitigate supervision dependence (Elskhawy et al., 2025; Dutta et al., 2025). Yet, they face two fundamental challenges. Figure 1 illustrates the core limitations of them and the motivation behind our approach. As shown in Figure 1 (a), most VLM-based SGG methods directly prompt a vision-language model to predict relational predicates between detected objects. While effective at recognizing salient entities, this flat inference

---
[1]School of Computer Science and Technology, Central South University, Hunan, China [2]School of Automation, Central South University,Hunan,China [3]School of Computer Science and Technology, Beijing Jiaotong University,Beijing, China. Correspondence to: Shichao Kan <kanshichao@csu.edu.cn>.

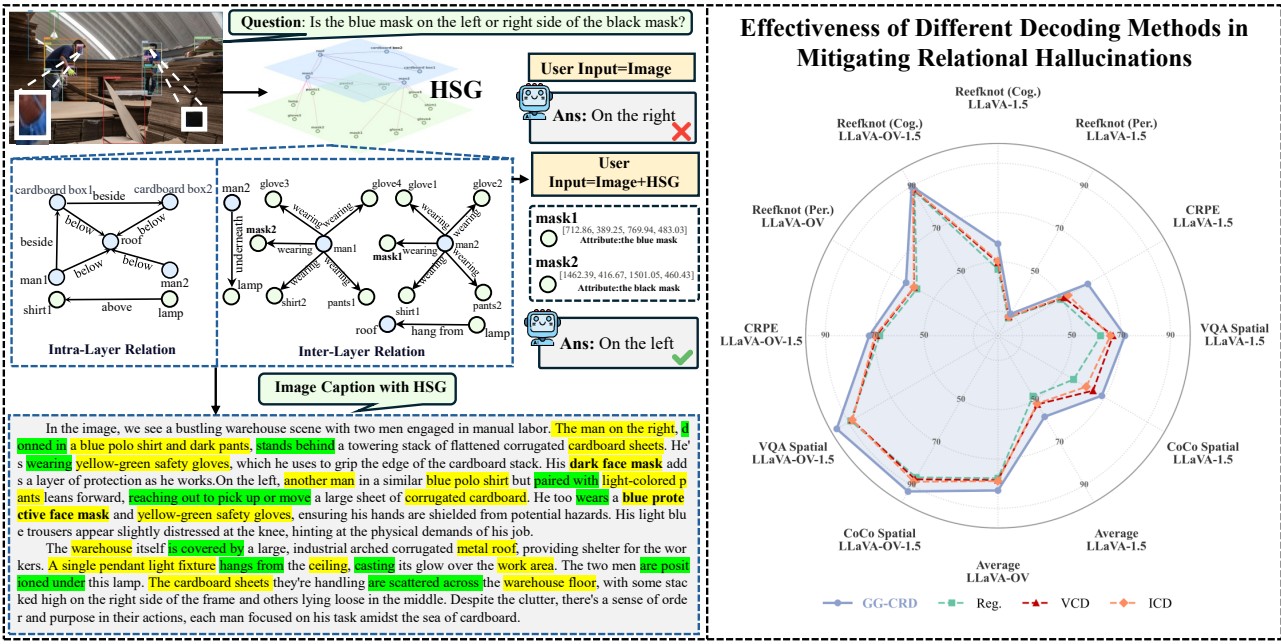

*Figure 2.* **Evaluating HSGG: Zero-Shot Hierarchical Scene Graphs and Enhanced Downstream Reasoning with Hallucination Mitigation. Left:** HSG generation and its effectiveness in downstream question answering and image captioning through explicit relational reasoning. **Right:** Comparison of decoding strategies in mitigating relational hallucinations across multiple benchmarks and model architectures.

paradigm lacks explicit structural and geometric constraints, causing the model to over-rely on semantic co-occurrence priors. As a result, visually implausible relations (e.g., predicting behind instead of in) often receive high confidence during decoding, leading to pervasive relational hallucinations in the generated scene graphs.

In contrast, our Hierarchical Scene Graph Generation (HSGG) framework, illustrated in Figure 1 (b), introduces a structured and geometry-aware inference pipeline. We first perform bidirectional hierarchical entity perception, where dominant objects are expanded top-down into constituent parts while fine-grained attributes are aggregated bottom-up, yielding a multi-level entity hierarchy that captures part–whole semantics and contextual dependencies. Building on these structured entities, relation inference is no longer performed over an unconstrained object set. Instead, we employ geometry-guided contrastive relation decoding, where a visually grounded expert model is contrasted against a hallucination-prone amateur model that relies only on geometric cues and linguistic priors. This contrastive formulation explicitly suppresses relations favored by spurious priors while preserving those consistent with both visual evidence and spatial geometry. Together, these two stages transform scene graph generation from unconstrained predicate decoding into a structured, geometry-grounded reasoning process.

We evaluate our approach from both structured relational

modeling and hallucination mitigation perspectives, as shown in Figure 2. On the left, HSGG generates fine-grained hierarchical scene graphs in open-world and zero-shot settings without training, explicitly modeling intra- and inter-layer relations and improving downstream visual question answering and image captioning. On the right, extensive experiments on Reefknot (Zheng et al., 2025), CRPE (Wang et al., 2024a), GQA-Spatial (Kamath et al., 2023), and COCO-Spatial (Kamath et al., 2023) demonstrate that GG-CRD consistently outperforms regular decoding as well as VCD (Leng et al., 2024) and ICD (Wang et al., 2024b) across two MLLM architectures, substantially reducing relational hallucinations. Together, these results show that structured relational representations and hallucination-aware decoding jointly enhance relational reasoning and reliability in multimodal models. Our contributions are summarized as follows:

- We formulate open-world hierarchical scene graph generation as a training-free reasoning problem, addressing two key limitations of existing VLM-based approaches: structural flatness and relational hallucinations.

- We propose bidirectional hierarchical entity perception for constructing multi-level scene hierarchies, and geometry-guided relation reasoning to prune implausible pairs and suppress hallucinated relations, enabling

semantically rich and spatially consistent scene graphs without additional training.

- Extensive experiments demonstrate that HSGG generalizes to unseen objects and predicates, reduces relational hallucinations, and consistently improves downstream reasoning tasks in open-world and zero-shot settings.

## 2. Related Work

**Scene Graph Generation (SGG).** SGG aims to parse an input image into a structured semantic graph, where nodes correspond to visual objects and their attributes, and edges encode the relationships between them (Krishna et al., 2017). Early SGG methods (Lu et al., 2016; Zellers et al., 2018; Tang et al., 2020; Chen et al., 2019) rely on large-scale manually annotated datasets for fully supervised training. While effective within the training domain, these approaches are limited by sparse annotations, fixed vocabularies, and poor generalization to open-world scenarios. To reduce annotation dependency, weakly-supervised SGG methods (Zareian et al., 2020; Kim et al., 2024) have been proposed, using only image-level labels to guide model learning. With the rapid progress of VLMs, Open-Vocabulary SGG (OVSGG) has gained attention. OVSGG aims to recognize object and predicate categories unseen during training. Current approaches (Chen et al., 2024b; Zhang et al., 2023) leverage pre-trained VLMs to extract vision-language aligned features for open-vocabulary concepts, but often require additional knowledge distillation or mapping to construct a relation space suitable for novel predicates. While these methods improve generalization, they still struggle with capturing multi-level scene hierarchies and compositional structures critical for fine-grained relational reasoning.

**Hallucination Mitigation in VLMs.** Despite their generative power, VLMs are prone to hallucinations, which can manifest as spurious object detections or incorrect relations. Existing mitigation strategies primarily focus on object-level errors, employing supervised paradigms such as Reinforcement Learning from Human Feedback (RLHF) (Dutta et al., 2025), hallucination-specific fine-tuning (Gunjal et al., 2024; Wu et al., 2025), or post-hoc revisors (Zhou et al., 2023). While effective, these approaches are computationally expensive and rely on non-scalable human annotations. To address these limitations, training-free inference-time strategies based on contrastive decoding have been proposed. Methods such as VCD (Leng et al., 2024) and ICD (Wang et al., 2024b) construct hallucination-prone amateur distributions via visual noise or instruction distortion, whereas SECOND (Park et al., 2025) employs selective multi-scale decoding to improve object grounding. However, these approaches primarily target object-level hallucinations and do not explicitly address relational hallucinations, leaving fine-grained predicate reasoning and geometry-aware relational grounding largely unresolved.

## 3. Method

### 3.1. Overview and Task Definition

We aim to parse an input image $I$ into a hierarchical scene graph $G = (\mathcal{V}, \mathcal{E})$, where each node $v_i \in \mathcal{V}$ represents an object with bounding box $b_i$, category $c_i$, and attributes $a_i$, and each edge $e_{ij} \in \mathcal{E}$ encodes a relational triplet $(c_i, r_{ij}, c_j)$. We focus on the *Open World* (OW) setting, where both object classes and predicate classes in the test phase can be novel with respect to the training set. This represents the most challenging setup, formally defined as $\forall (o_i, o_j, r_{ij}) \in \mathcal{E}_{\text{test}}, \quad (o_i, o_j) \notin \mathcal{O}_{\text{train}} \land r_{ij} \notin \mathcal{R}_{\text{train}}$. Scene graph generation in open-world settings presents several challenges: (1) the long-tailed distribution of relations and limited annotations make it difficult to generalize to unseen objects or predicates; (2) traditional SGG approaches produce flat scene graphs that fail to capture part–whole hierarchies; and (3) vision-language models often hallucinate relational predicates due to co-occurrence biases, leading to spurious edges. These challenges complicate zero-shot and open-world reasoning, necessitating both hierarchical node modeling and geometry-aware relation validation.

To address these challenges, we propose HSGG, a fully training-free framework for open-world hierarchical scene graph generation, which is organized into two sequential stages, as illustrated in Figure 3. In the first stage (Figure 3 (a)), bidirectional hierarchical entity perception (BHEP) constructs multi-level scene hierarchies by integrating top-down object expansion with bottom-up fine-grained attribute reasoning, enabling structured entity representations that capture part–whole semantics. In the second stage (Figure 3 (b)), geometry-guided relation reasoning operates on these structured entities to infer reliable relations. This stage consists of two tightly coupled components: geometry-aware relation filtering (GA-RF) (Figure 3 (b)-(1)), which prunes spatially implausible object pairs using geometric cues, and geometry-grounded contrastive relation decoding (GG-CRD) (Figure 3 (b)-(2)), which suppresses hallucinated predicates by contrasting visually grounded predictions with geometry-induced priors. Together, these two stages produce a coherent hierarchical scene graph that supports robust open-world reasoning. In the following subsections, we present each component in detail.

### 3.2. Bidirectional Hierarchical Entity Perception (BHEP)

A major limitation of existing SGG approaches is their reliance on flat object representations, which fail to capture the intrinsic compositionality, part–whole hierarchies, and

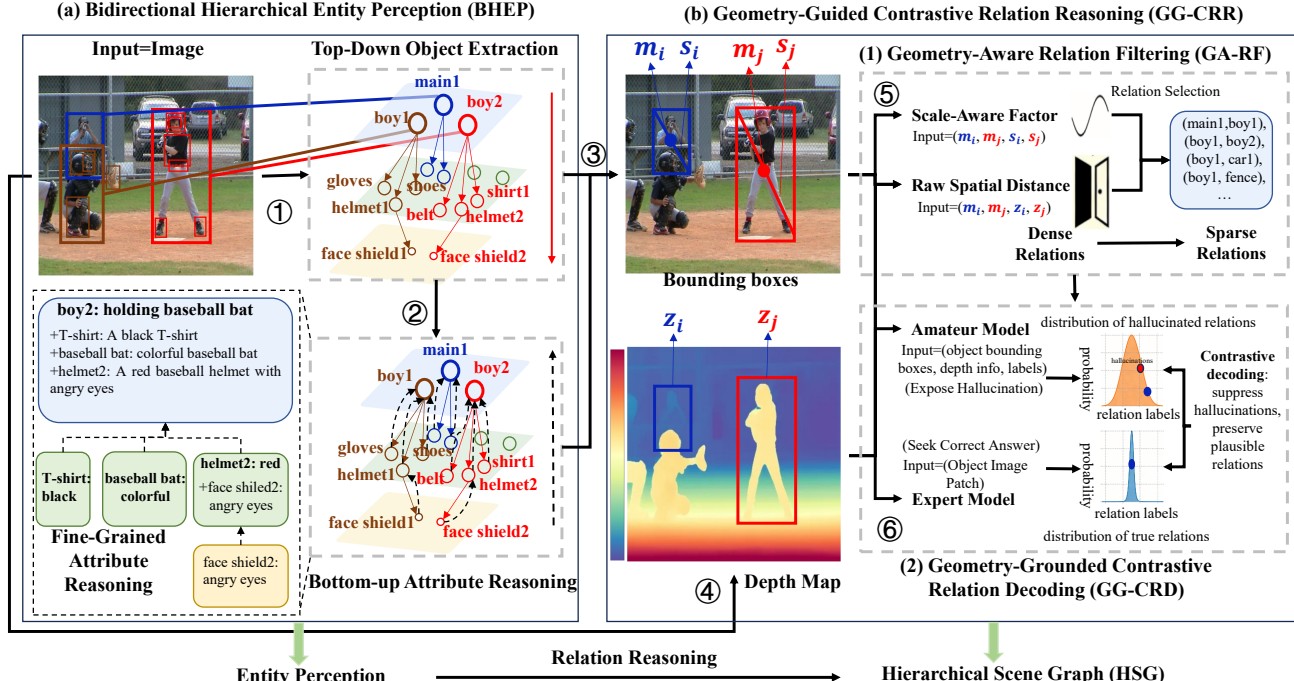

*Figure 3.* Technical pipeline of HSGG. (a) Bidirectional hierarchical entity perception builds hierarchical entities by integrating top-down object extraction with bottom-up fine-grained attribute reasoning. (b) Geometry-guided contrastive relation reasoning performs relation filtering and decoding under geometric constraints to generate the hierarchical scene graph

contextual dependencies present in real-world scenes. Such flat representations hinder fine-grained relational reasoning and reduce robustness in open-world or zero-shot settings, where unseen objects and complex structures are common. To address these challenges, we propose a bidirectional hierarchical reasoning strategy that simultaneously models top-down and bottom-up flows, as shown in Figure 3 (a):

**Top-Down Object Extraction:** We first identify salient objects as root nodes using a VLM. Each root node is then iteratively expanded to discover constituent parts and contextual sub-entities, forming a hierarchical tree structure. Formally, at hierarchy level $\ell$, a node $v_i^\ell$ is represented as a label–box pair $(o_i^\ell, b_i^\ell)$, and its children are generated via:

$$\{o_k^{\ell+1}, \ldots, o_{k+n}^{\ell+1}\} = M\big(I, P(o_i^\ell)\big), \quad (1)$$

where $M(\cdot)$ is the VLM and $P(\cdot)$ constructs a parent-conditioned prompt. Each child is grounded to the image using an open-vocabulary detector $\mathcal{D}(\cdot)$, ensuring accurate spatial localization. This top-down process captures global-to-local object dependencies and discovers multi-level scene structures.

**Bottom-Up Attribute Reasoning:** While the top-down pass provides the hierarchical skeleton of objects, it does not fully capture fine-grained semantic details, such as attributes, which are often sparsely annotated in datasets but crucial for relational reasoning. To address this, we perform

bottom-up attribute inference: leaf nodes are first processed to predict attributes from localized image patches, and these attributes are propagated upward to parent nodes, fusing visual evidence from children with parent-level cues:

$$a_j^\ell = \mathcal{M}\Big(I_j^\ell, \mathcal{P}(\{a_k^{\ell+1} \mid v_k^{\ell+1} \in \text{Child}(v_j^\ell)\})\Big), \quad (2)$$

where $M(\cdot)$ is the VLM, $I_j^\ell$ is the visual patch of node $v_j^\ell$, and $\mathcal{P}(\cdot)$ encodes child attributes as textual prompts.

By combining top-down object expansion with bottom-up attribute reasoning, BHEP captures both the hierarchical structure of the scene and fine-grained semantic context, producing rich node representations that support accurate relation inference and hierarchical scene understanding.

### 3.3. Geometry-Guided Contrastive Relation Reasoning (GG-CRR)

After hierarchical nodes are extracted via BHEP, each object is associated with a bounding box, category, and attributes, and the next challenge is to infer relations between these nodes. Considering all object pairs naively leads to quadratic growth of candidate relations, many of which are spatially implausible, while VLMs are prone to hallucinate relational predicates due to co-occurrence biases. To address both challenges, we introduce a geometry-aware contrastive relation reasoning strategy that first prunes can-

didates based on spatial constraints and then suppresses hallucinated relations via contrastive relation decoding, as shown in Figure 3 (b). In the following, we introduce each of them specifically.

**Geometry-Aware Relation Filtering (GA-RF).** After hierarchical entity extraction via BHEP, each node is associated with a bounding box, category, and attributes. Inferring all possible object pairs for relations leads to quadratic growth in candidates, many of which are geometrically implausible. Simple heuristics based on 2D proximity or bounding-box overlap can prune some pairs, but they often fail to capture real-world interactions, for example, large objects may interact over long distances, and depth cues are typically ignored. To address these challenges, we propose a geometry-aware relation filtering strategy that integrates 2D spatial proximity, 3D depth, and object-scale awareness to efficiently reduce the candidate relation set.

For a candidate pair $(o_i, o_j)$ with 2D centers $m_i, m_j$, bounding-box diagonals $s_i, s_j$, and median depths $z_i, z_j$, we first compute a raw spatial distance that penalizes pairs separated in the image plane or in depth:

$$d_{\text{raw}} = \underbrace{\frac{\|m_i - m_j\|_2}{\sqrt{H^2 + W^2}}}_{\text{2D center-to-center proximity, penalizing distant objects}} + \underbrace{\frac{|z_i - z_j|}{\Delta Z}}_{\text{3D depth difference, penalizing objects at very different depths}}, \quad (3)$$

where $H, W$ are the image height and width, and $\Delta Z$ is the scene depth range. This metric captures both image-plane and depth separation but ignores object size. To avoid penalizing large objects that naturally span long distances, we introduce a scale-aware factor:

$$\psi_{ij} = \frac{\delta_{ij}}{\delta_{ij} + \frac{1}{2}(s_i + s_j)}, \quad \delta_{ij} = \|m_i - m_j\|_2, \quad (4)$$

where $s_i, s_j$ are the diagonals of the object bounding boxes. The final pruning score is then

$$d_{\text{final}} = d_{\text{raw}} \cdot \psi_{ij}, \quad (5)$$

and a pair is retained only if $d_{\text{final}} < \tau$. This ensures that large objects interacting over long distances are preserved, while geometrically implausible pairs are discarded, significantly reducing the relation search space for downstream reasoning.

**Geometry-Grounded Contrastive Relation Decoding (GG-CRD).** Even after pruning implausible object pairs, VLMs can still predict spurious relations due to semantic co-occurrence biases and internal priors. To address this, we introduce geometry-grounded contrastive relation decoding, which leverages two complementary models with distinct

roles: an expert model that has access to full visual context to produce reliable relation predictions, and a hallucination-prone amateur model that sees only geometric cues and linguistic priors to simulate likely spurious relations. Comparing the outputs of these two models allows the system to suppress false relations while preserving semantically and spatially plausible ones.

For a candidate object pair $(o_i, o_j)$, the expert model receives the cropped image patch $\mathbf{I}_{ij}$ along with a relation query prompt $\mathbf{x}_q$ and generates a token distribution at each decoding step:

$$L_{\text{ex}} = \text{Logit}_\theta(y_t \mid y_{<t}, \mathbf{I}_{mn}, \mathbf{x}_q), \quad (6)$$

where $y_{<t}$ represents previously generated tokens. This distribution captures the model's grounded prediction based on full visual context. The amateur model, in contrast, is provided only with geometric information $\mathbf{x}_{\text{geom}} = \{b_i, b_j, d_i, d_j\}$, where $b_i$ and $b_j$ denote the 2D bounding boxes of the subject and object, and $d_i$ and $d_j$ represent their corresponding relative depth estimates. Together with the same linguistic priors, it produces:

$$L_{\text{am}} = \text{Logit}_\theta(y_t \mid y_{<t}, \mathbf{x}_{\text{geom}}, \mathbf{x}_q), \quad (7)$$

thereby simulating the distribution of hallucinated relations that arise from spatial biases and co-occurrence priors.

To ensure that low-confidence tokens from the expert do not dominate the contrastive process, we introduce an adaptive plausibility constraint. First, we compute the softmax probability of expert predictions:

$$P_{\text{ex}} = \text{softmax}(L_{\text{ex}}),$$

and define a truncated candidate set containing only sufficiently probable tokens:

$$\mathcal{V}_{\text{apc}} = \{y_t \in \mathcal{V} \mid P_{\text{ex}}(y_t) \geq \beta \cdot \max_{w \in \mathcal{V}} P_{\text{ex}}(w)\}, \quad (8)$$

where $\beta \in [0, 1]$ is a hyperparameter controlling the truncation threshold. Finally, the contrastive token selection combines expert and amateur predictions while respecting this plausibility mask:

$$y_t \sim \text{softmax}\left( \begin{cases} L_{\text{ex}} + \alpha(L_{\text{ex}} - L_{\text{am}}), & y_t \in \mathcal{V}_{\text{apc}}, \\ -\infty, & \text{otherwise.} \end{cases} \right) \quad (9)$$

This unified approach ensures that relation predictions are both geometrically grounded and semantically coherent, effectively suppressing hallucinated relations while preserving valid interactions dictated by the scene layout.

---

**Algorithm 1** HSGG: Open-World Hierarchical Scene Graph Generation

---

1: **Input:** Image $I$, threshold $\tau$, ,contrastive factor $\alpha$ ,truncation threshold $\beta$
2: **Output:** Hierarchical Scene Graph $G = (\mathcal{V}, \mathcal{E})$
   **Step 1: Bidirectional Hierarchical Entity Perception (BHEP)**
3: $\mathcal{V} \leftarrow \text{TopDownObjectExtraction}(I)$ ▷ extract hierarchical object candidates
4: **for** each node $v_i \in \mathcal{V}$ **do**
5:   $a_i \leftarrow \text{BottomUpAttributeReasoning}(I, v_i)$ ▷ propagate attributes from parts to wholes
6: **end for**
   **Step 2: Geometry-Guided Contrastive Relation Reasoning**
7: $\mathcal{E} \leftarrow \emptyset, \quad Z \leftarrow \text{DepthEstimation}(I)$ ▷ initialize relations and depth cues
8: **for** each candidate pair $(v_i, v_j) \in \mathcal{V} \times \mathcal{V}$ **do**
9:   $d_{\text{final}} \leftarrow \text{GA-RF}(b_i, b_j, z_i, z_j)$ ▷ geometry-aware relation filtering
10:   **if** $d_{\text{final}} < \tau$ **then**
11:     $L_{\text{ex}} \leftarrow \text{Logit}_\theta(y_t \mid I_{ij}, \mathbf{x}_q)$ ▷ expert prediction with visual grounding
12:     $L_{\text{am}} \leftarrow \text{Logit}_\theta(y_t \mid \mathbf{x}_{\text{geom}}, \mathbf{x}_q)$ ▷ hallucination-prone geometric prior
13:     $\mathcal{V}_{\text{apc}} \leftarrow \{y_t \mid P_{\text{ex}}(y_t) \geq \beta \cdot \max P_{\text{ex}}\}$ ▷ adaptive plausibility constraint
14:     $r_{ij} \leftarrow y_t \sim \text{softmax}\big(\{L_{\text{ex}} + \alpha(L_{\text{ex}} - L_{\text{am}}) \mid y_t \in \mathcal{V}_{\text{apc}}\}\big)$ ▷ geometry-grounded contrastive relation decoding
15:     $\mathcal{E} \leftarrow \mathcal{E} \cup \{(c_i, r_{ij}, c_j)\}$ ▷ add relation edge
16:   **end if**
17: **end for**
18: **return** $G = (\mathcal{V}, \mathcal{E})$

---

## 3.4. Scene Graph Generation Procedure

As shown in **Algorith 1**, the inference pipeline of HSGG follows a structured, training-free procedure that decouples entity perception from relation reasoning. Given an input image $I$, HSGG first establishes a hierarchical entity representation that serves as the structural foundation of the scene, and then infers relations under explicit geometric constraints. Specifically, the pipeline begins with BHEP to construct the node set $\mathcal{V}$. BHEP performs a bidirectional reasoning process over the image: a top-down expansion stage leverages a VLM to identify dominant objects and recursively decompose them into constituent sub-parts, forming a hierarchical object skeleton; this is followed by a bottom-up attribute reasoning stage, where fine-grained semantic attributes inferred from localized image patches are propagated from leaf nodes to higher-level entities. Through this bidirectional interaction, BHEP produces a structured

set of nodes $\mathcal{V} = \{v_i\}$, each associated with a bounding box $b_i$, a category label $c_i$, and a set of semantic attributes $a_i$, yielding a rich and hierarchical representation of scene entities.

Building upon the extracted entities, HSGG constructs the final scene graph via geometry-guided relation reasoning. The GA-RF module first examines all possible node pairs in $\mathcal{V}$ and prunes geometrically implausible interactions based on 2D spatial proximity, 3D depth consistency, and object-scale awareness. This step substantially reduces the quadratic relation search space to a sparse candidate set $\mathcal{E}_{\text{cand}}$ by retaining only pairs that satisfy a geometric threshold $\tau$. For each remaining candidate pair, the GG-CRD module performs contrastive inference by comparing the predictions of a visually grounded expert model, which has access to the cropped image region, against a geometry-biased amateur model that relies only on degraded geometric cues and linguistic priors. By suppressing predicates favored by geometric or co-occurrence biases while amplifying visually consistent signals, GG-CRD effectively mitigates relational hallucinations and yields reliable predicates $r_{ij}$. Finally, the validated relational triplets are integrated with the hierarchical node set to form the output scene graph $G = (\mathcal{V}, \mathcal{E})$. The entire pipeline operates without any task-specific training, enabling robust zero-shot and open-world generalization while producing fine-grained, spatially grounded hierarchical scene graphs.

## 4. Experiment

Scene graph benchmarks often contain noisy, incomplete, or biased labels, making standard SGG metrics insufficient to fully capture structural completeness and relational fidelity. To provide a more comprehensive assessment, we evaluate HSGG on both conventional SGG benchmarks and downstream reasoning tasks, examining relation accuracy and the utility of generated graphs in high-level visual reasoning.

**Datasets.** We evaluate HSGG on Visual Genome (VG) (Krishna et al., 2017) and Panoptic Scene Graph Generation (PSG) (Yang et al., 2022) for standard SGG, and on CompreCap (Lu et al., 2025) and V*Bench (Wu & Xie, 2024) for image captioning and fine-grained visual question answering. To test our geometry-guided relation reasoning module against relational hallucinations (Figure 2), we also include four spatial reasoning benchmarks: Reefknot (Zheng et al., 2025), CRPE (Wang et al., 2024a), GQA-Spatial (Kamath et al., 2023), and COCO-Spatial (Kamath et al., 2023). More details about the datasets are provided in Appendix B

**Evaluation Metrics.** For scene graph generation, we follow the standard SGDet evaluation protocol and report Recall@K (R@K) and mean Recall@K (mR@K). For all downstream tasks, including image captioning and visual

question answering, we report accuracy as the primary evaluation metric, ensuring a consistent and interpretable comparison across different tasks.

## 4.1. Comparison with State-of-the-art SGG Methods

We evaluate HSGG on two challenging settings, i.e., *zero-shot SGG* and *open-world SGG*, to assess generalization beyond closed-vocabulary supervision. Zero-shot SGG evaluates unseen relation triplets with known categories, while open-world SGG further requires generalization to unseen object and predicate categories at inference time. Experiments are conducted on the VG and PSG datasets following the standard SGDet protocol, with Recall@K (R@K) as the primary metric.

**Zero-Shot Scene Graph Generation.** We first evaluate HSGG under the zero-shot setting using the VG150 and standard PSG protocols, comparing against recent state-of-the-art methods including OWSGG, PGSG, and CAPSGG. To address the mismatch between HSGG's open-vocabulary predictions and benchmark label spaces, we apply a semantic vocabulary alignment strategy based on Qwen3-Embedding and cosine similarity. As shown in Table 1, HSGG achieves state-of-the-art performance on both datasets. On VG, it outperforms the strongest training-free baseline OWSGG by +12.3% / +13.1% (R@50/100) and surpasses the trained CAPSGG by +1.0% / +1.8%. On PSG, HSGG improves over OWSGG by +6.7% / +7.7% and exceeds the trained PGSG by +4.6% / +5.2%, demonstrating strong zero-shot generalization without task-specific training.

**Open-World Scene Graph Generation.** We further evaluate HSGG in the open-world SGG setting, where both object and predicate categories are unseen at test time. As shown in Table 2, HSGG outperforms the previous best training-free method OWSGG by +7.4% / +9.3% (R@50/100), and surpasses the state-of-the-art trained model OVSGTR by +3.3% / +1.9%. These results demonstrate the robustness of HSGG under realistic open-world conditions, even without any task-specific training.

## 4.2. Downstream Evaluation on Visual Question Answering

We further evaluate the transferability of HSGG-generated scene graphs on visual question answering using the $V^*$ Bench dataset, which focuses on attribute recognition and fine-grained spatial reasoning—two known challenges for VLMs relying on implicit visual representations.

As shown in Table 3, image-only baselines exhibit clear limitations on fine-grained VQA, with most models achieving accuracies below 75%. In contrast, incorporating HSGG-generated scene graphs leads to consistent and significant

*Table 1.* Experimental results of zero-shot setting on VG and PSG test set.

| Method | Dataset | Zero-Shot | | |
|---|---|---|---|---|
| | | zR@20 | zR@50 | zR@100 |
| OwSGG (Dutta et al., 2025) | VG150 | 0.7 | 1.2 | 2.0 |
| RGN (Suhail et al., 2021) | | 1.7 | 3.2 | 5.3 |
| SSRCNN (Teng & Wang, 2022) | | - | 3.1 | 4.5 |
| SGTR (Li et al., 2022) | | - | 2.5 | 5.8 |
| PGSG (Li et al., 2024b) | | - | 6.2 | 8.5 |
| T-CAR (Li et al., 2023b) | | 3.6 | 6.2 | 8.7 |
| CAPSGG (Huang et al., 2025) | | 9.3 | 12.5 | 13.3 |
| **HSGG (Ours)** | | **10.1** | **13.5** | **15.1** |
| PSGTR (Yang et al., 2022) | PSG | - | 3.1 | 6.4 |
| SGTR (Li et al., 2022) | | - | 4.1 | 5.8 |
| OwSGG (Dutta et al., 2025) | | 2.8 | 4.7 | 6.4 |
| PGSG (Li et al., 2024b) | | - | 6.8 | 8.9 |
| **HSGG (Ours)** | | **7.2** | **11.4** | **14.1** |

*Table 2.* Experimental results of Open-World setting on VG dataset.

| Method | Open-World | | |
|---|---|---|---|
| | R@20 | R@50 | R@100 |
| OwSGG(Qwen) (Dutta et al., 2025) | - | 1.6 | 2.4 |
| OwSGG(LLaVA-Next) (Dutta et al., 2025) | - | 1.9 | 2.6 |
| OvSGTR (Swin-B) (Chen et al., 2024b) | - | 6.0 | 10.0 |
| **HSGG (Ours)** | **6.4** | **9.3** | **11.9** |

improvements across models. For InternVL3-VL-8B, attribute and spatial accuracies increase from 72.17% to 74.78% and from 71.05% to 78.95%, respectively, yielding a 4.7% gain in overall accuracy. Similarly, LLaVA-OV-1.5-8B benefits from scene graph integration, improving attribute and spatial reasoning by 2.6% and 2.63% and achieving the best overall performance among all methods. These results indicate that HSGG-generated scene graphs act as an explicit intermediate representation that externalizes object attributes and spatial relations, thereby alleviating the need for implicit structure inference and enabling more reliable and interpretable visual reasoning in downstream VQA tasks.

## 4.3. Downstream Evaluation on Image Captioning.

We evaluate the effectiveness of HSGG-generated scene graphs on image captioning to assess how structured visual representations support multimodal generation. Two prompting settings are considered: (1) image-only input and (2) scene-graph-only input, where the model accesses an explicit structured abstraction without raw visual features. Instead of conventional captioning metrics that weakly reflect relational correctness, we adopt the CompreCap protocol, which evaluates captions along four interpretable dimensions—*object*, *attribute*, *relation*, and *holistic*. This enables a fine-grained and structure-aware assessment aligned with the compositional semantics encoded in scene graphs.

As shown in Table 4, image-only baselines benefit from increased model scale, but the gains are mainly reflected in object coverage and caption length, with limited improvement in attribute and relation modeling. This indicates that scaling

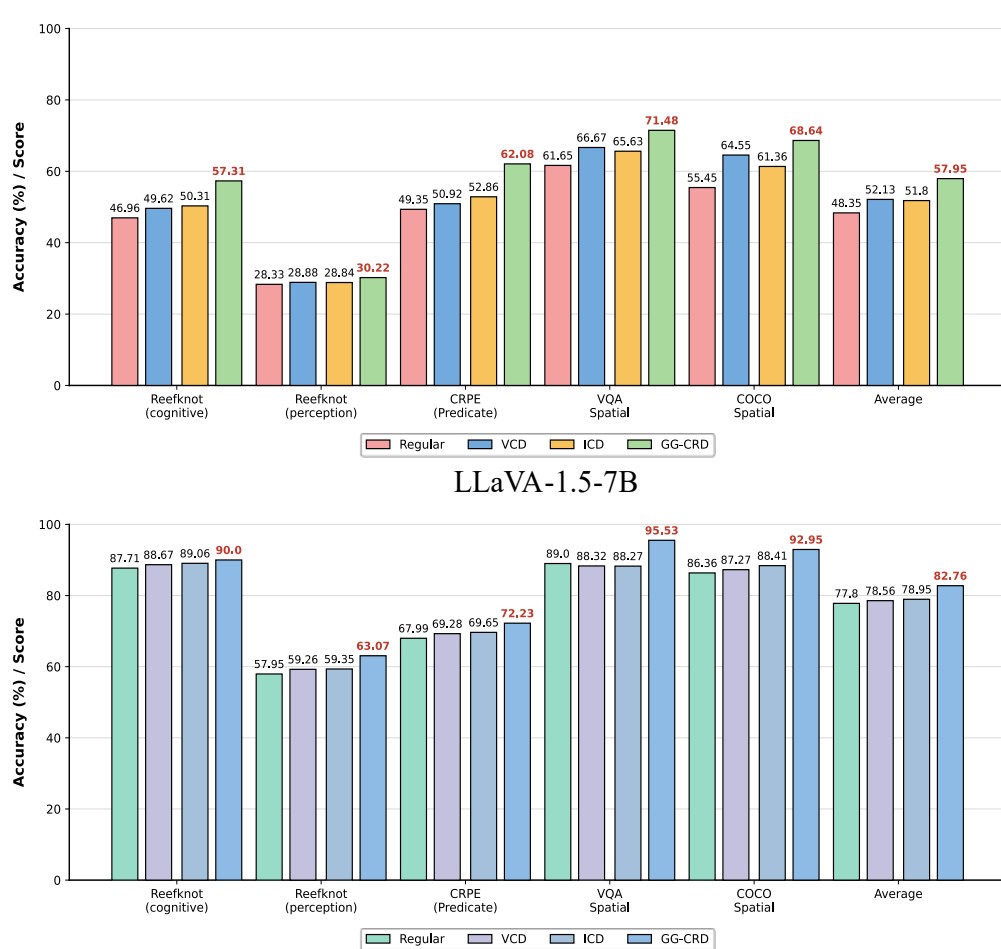

*Figure 4.* Performance comparison of various decoding strategies across multiple relational hallucination benchmarks.

*Table 3.* Downstream Evaluation on Visual Question Answer.

| Model | Attribute (%) | Spatial (%) | Overall (%) |
|---|---|---|---|
| InstructBLIP(Dai et al., 2023) | 25.21 | 47.36 | 34.02 |
| MiniGPT-4(Zhu et al., 2023) | 30.43 | 50.00 | 38.22 |
| BLIP2 (Li et al., 2023a) | 26.95 | 53.94 | 37.69 |
| LLaVA-1.5 (Liu et al., 2024) | 43.47 | 56.57 | 48.68 |
| Qwen2.5-VL-7B (Bai et al., 2025) | 74.78 | 73.68 | 74.34 |
| InternVL3-8B (Zhu et al., 2025) | 72.17 | 71.05 | 71.73 |
| LLaVA-OV-1.5-8B (An et al., 2025) | 79.13 | 73.68 | 76.96 |
| InternVL3-8B+HSG | 74.78 | **78.95** | 76.43 |
| LLaVA-OV-1.5-8B+HSG | **81.73** | 76.31 | **79.58** |

alone does not consistently enhance fine-grained semantic composition. In contrast, using HSGG-generated scene graphs as the sole input yields consistent improvements across all CompreCap dimensions. Specifically, LLaVA-OV-1.5-8B with scene-graph-only input improves the unified score by 3.27% over its image-only counterpart, with notable gains in object accuracy and relation recognition. Despite the absence of visual features, relation scores remain competitive, demonstrating that explicit relational structures effectively preserve interaction-level semantics. Moreover, this 8B model outperforms substantially larger image-based models, including LLaVA-Next-34B and GPT-4o, in unified

score. These results suggest that HSGG-generated scene graphs serve as a strong inductive bias, enabling compact models to achieve more faithful and compositional caption generation than raw pixel-level representations.

### 4.4. Effectiveness in Mitigating Relational Hallucinations

To evaluate the effectiveness of GG-CRD in reducing relational hallucinations, we conduct extensive experiments on the CRPE, ReefNot, COCO-Spatial, and GQA-Spatial benchmarks. We compare GG-CRD with multiple decoding strategies across different model architectures, with the main results summarized in Figure 4. Across all benchmarks, GG-CRD achieves the strongest performance, indicating its robustness in preserving relational factuality during text generation.

Quantitative results further demonstrate the advantage of GG-CRD under different decoding settings. When applied to LLaVA-OneVision-1.5-7B, GG-CRD yields an average improvement of 4.96% over regular decoding, outperform-

*Table 4.* Downstream Evaluation on Image Caption

| Model | Caption Length | $S_{object}$ (%) ↑ | $S_{attribute}$ (0∼5) ↑ | $S_{relation}$ (0∼5) ↑ | S-Cov (%) ↑ | $S_{unified}$ (0∼100) ↑ |
|---|---|---|---|---|---|---|
| LLaVA-1.5-13B (Liu et al., 2024) | 86.97 | 59.86 | 2.01 | 2.59 | 43.81 | 50.32 |
| ShareGPT4V-13B (Chen et al., 2024a) | 155.91 | 67.88 | 2.40 | 2.69 | 55.86 | 55.56 |
| LLaVA-Next-Llama3-8B (Li et al., 2024a) | 168.99 | 70.22 | 2.48 | 2.72 | 56.95 | 56.91 |
| miniGemini-HD-34B (Li et al., 2024c) | 173.71 | 70.70 | 2.48 | 2.70 | 57.20 | 56.88 |
| InternVL3-8B (Zhu et al., 2025) | 125.68 | 75.56 | 2.67 | 2.63 | 56.91 | 58.62 |
| Qwen2.5-VL-7B (Bai et al., 2025) | 133.69 | 75.70 | 2.63 | 2.61 | 57.49 | 57.22 |
| LLaVA-Next-34B (Li et al., 2024a) | 179.24 | 72.86 | 2.59 | 2.79 | 58.49 | 58.85 |
| GPT-4o (Islam & Moushi, 2025) | 108.20 | 72.78 | 2.58 | **2.93** | 57.54 | 60.05 |
| LLaVA-OV-1.5-8B (Input=Image) | 123.71 | 78.19 | 2.64 | 2.71 | 58.87 | 59.70 |
| LLaVA-OV-1.5-8B (Input=HSG) | 204.36 | **83.04** | **2.75** | 2.87 | **60.15** | **62.97** |

*Table 5.* Ablation study on Component Contribution.

| BHEP | GA-RF | GG-CRD | zR@20 | zR@50 | zR@100 |
|---|---|---|---|---|---|
| | | ✓ | 5.23 | 6.89 | 7.35 |
| | ✓ | ✓ | 5.39 | 7.12 | 7.58 |
| ✓ | | | 6.53 | 9.82 | 11.63 |
| ✓ | ✓ | | 6.92 | 10.73 | 12.13 |
| ✓ | ✓ | ✓ | 7.18 | 11.42 | 14.05 |

ing VCD and ICD by 4.20% and 3.81%, respectively. Similar gains are observed on LLaVA-1.5-7B, where GG-CRD improves performance by 9.60% relative to regular decoding, exceeding VCD and ICD by 5.82% and 6.15% points. These consistent improvements across models and benchmarks highlight the effectiveness of GG-CRD in mitigating relational hallucinations in vision-language generation.

### 4.5. Ablation Study

**Component Contribution Analysis.** We analyze the contribution of each HSGG component: Bidirectional Hierarchical Entity Perception (BHEP), Geometry-Aware Relation Filtering (GA-RF), and Geometry-Grounded Contrastive Relation Decoding (GG-CRD) on panoptic Scene Graph Generation datasets in zero-shot setting. As shown in Table 5, GG-CRD alone performs poorly in zero-shot recall, indicating that relation refinement is ineffective without sufficient entity coverage, and adding GA-RF yields only marginal gains under this constraint. In contrast, introducing BHEP leads to a substantial improvement across all recall metrics by enabling comprehensive entity discovery. Building upon BHEP, GA-RF further improves ZR@50/100 by reducing redundant object pairs and relational noise. Finally, integrating GG-CRD into the full pipeline provides an additional 1.54%/1.92% gain on ZR@50/100, confirming its effectiveness in suppressing low-confidence and implausible triplets at high-recall regimes. Overall, the ablation results reveal a clear functional hierarchy: BHEP expands recall, GA-RF controls relational complexity, and GG-CRD refines precision.

**Effect of Relation Filtering Threshold.** We further examine the sensitivity of the GA-RF module to the filter-ing threshold $\tau$. Based on a GPT-4o–annotated subset of the PSG dataset, we evaluate the precision–recall trade-off across different $\tau$ values. Empirically, $\tau = 0.3$ achieves the best balance between retaining plausible relational candidates and suppressing spurious pairs, and is therefore adopted in all experiments. Detailed analysis is provided in Appendix C.5.

## 5. Conclusion

We have presented HSGG, a fully training-free framework for open-world hierarchical scene graph generation, which explicitly decouples entity perception from relation reasoning. By integrating bidirectional hierarchical entity perception with geometry-guided relation reasoning, HSGG constructs multi-level scene hierarchies and infers relations that are both semantically coherent and geometrically grounded. Extensive experiments demonstrate that each component contributes meaningfully: hierarchical entity perception expands candidate objects, relation filtering reduces combinatorial noise, and contrastive relation decoding suppresses hallucinated predicates. HSGG achieves state-of-the-art performance in zero-shot and open-world SGG benchmarks, while also improving downstream tasks such as visual question answering and image captioning, even when only the generated scene graph is provided. These results highlight that structured, hierarchical scene representations can serve as a powerful inductive bias, enabling robust relational reasoning and fine-grained multimodal understanding in open-world scenarios.

## Acknowledgments

This work was supported in part by the State Key Program of National Natural Science Foundation of China (Grant No. 62233018), in part by the National Natural Science Foundation of China under Grant 62473033, 62463002, and 62473384, in part by the Beijing Natural Science Foundation under grant L231012. We are grateful to the High Performance Computing Center of Central South University for partial support of this work.

## Impact Statement

This paper presents work whose goal is to advance the field of Machine Learning. There are many potential societal consequences of our work, none which we feel must be specifically highlighted here.

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

# A. Implementation Details

We adopt LLaVA-OneVision-1.5-8B (An et al., 2025) as the core vision-language model for entity perception, attribute recognition, and relational reasoning. Open-vocabulary object localization is performed using LLMDet (Fu et al., 2025), which provides high-quality bounding boxes for hierarchical entity construction. To incorporate geometric cues, we employ Depth Anything V3 (Lin et al., 2025) to estimate dense depth maps from input images.To facilitate more effective evaluation in the open-vocabulary setting, we adopt Qwen3-Embedding-4B for semantic alignment. Furthermore, we use Qwen3-VL-4B to evaluate semantic consistency.For hierarchical entity perception, the maximum hierarchy depth is set to $\ell = 3$, and the geometry-aware relation filtering threshold is fixed to $\tau = 0.3$. In GG-CRD, we set the contrastive factor $\alpha = 1$ and truncation threshold $\beta = 0.1$ across all experiments.

# B. Benchmark Detail

### B.1. SGG Benchmarks

**Visual Genome (VG)** (Krishna et al., 2017) is the most prevalent benchmark in the SGG field, containing approximately 108k images with dense annotations of objects and their pairwise relationships. Due to the significant noise and long-tail distribution in the original annotations, we follow the widely-adopted preprocessing protocol established by (Xu et al., 2017). This refined version, commonly referred to as VG150, focuses on the most frequent 150 object categories and 50 predicate classes. Regarding the zero-shot setting, we follow the protocol established in (Huang et al., 2025) to evaluate performance exclusively on unseen triplets. For the more challenging open-world setting, we adopt the evaluation strategy from (Dutta et al., 2025), focusing on triplets where both the object categories and predicates are entirely unseen during training to assess the model's generalization in unconstrained environments.

**Panoptic Scene Graph (PSG)** (Yang et al., 2022) is a more recent and challenging benchmark. Unlike VG, which uses bounding boxes that may overlap or be ambiguous, PSG requires models to ground relationships on precise segments. It comprises 49k images covering 133 object classes and 56 predicate categories. This dataset is particularly suited for evaluating a model's ability to capture fine-grained spatial and semantic dependencies.Regarding the zero-shot setting, we follow the protocol established in (Shi et al., 2025) to evaluate performance exclusively on unseen triplets.

### B.2. Downstream Task Benchmarks

**V*Bench** (Wu & Xie, 2024) consists of 191 curated high-resolution images paired with complex queries, specifically designed to evaluate fine-grained attribute recognition and spatial reasoning for small objects. This characteristic makes V*Bench particularly well-suited for the evaluation of Hierarchical Scene Graphs (HSG). Given that HSG effectively captures minute visual instances through its hierarchical perception mechanism, V*Bench provides an ideal testbed to precisely validate the advantages of HSG in parsing fine-grained object attributes and capturing complex spatial dependencies.

**CompreCap** (Lu et al., 2025) is a benchmark designed to evaluate the information density and completeness of image captions. Unlike standard datasets that favor concise summaries, CompreCap provides dense annotations covering object counts, fine-grained attributes, and complex inter-entity relationships. This dataset is uniquely suited for testing Hierarchical Scene Graphs (HSG), as its evaluation demands a structural understanding of scenes from the global context down to localized details. By employing HSG, the model can systematically traverse different semantic levels to generate exhaustive descriptions without missing minute entities. Consequently, CompreCap provides a rigorous platform to verify whether HSG can effectively organize high-density visual information into coherent, multi-faceted narratives.

### B.3. Relational Hallucination Benchmarks

To rigorously evaluate the efficacy of Geometry-Grounded Contrastive Relation Decoding (GG-CRD) in mitigating relational hallucinations, we conduct extensive experiments on four specialized benchmarks: Reefknot(Zheng et al., 2025), CRPE (Wang et al., 2024a), GQA-Spatial, and COCO-Spatial(Kamath et al., 2023). For Reefknot, we report performance across both perception and cognition tasks to provide a multi-dimensional assessment of visual reasoning. Regarding CRPE, we specifically focus our evaluation on the more challenging Predicate Category subset, which demands precise discrimination of subtle relation transitions. Collectively, these benchmarks serve as a stringent testbed to verify whether our GG-CRD module can effectively rectify hallucinated spatial dependencies and enhance relational fidelity through GG-CRD.

*Table 6.* Proportions of hallucinated (H) and correct (C) relations across amateur-score bins, where H indicates hallucinated relations and C indicates correct relations.

| score bin | Reefnot_Cog (H) | Reefnot_Cog (C) | Reefnot_Per (H) | Reefnot_Per (C) |
|---|---|---|---|---|
| low-score | 9.09% | 90.91% | 10.05% | 89.95% |
| medium-score | 76.62% | 23.38% | 74.13% | 25.87% |
| high-score | 94.03% | 5.97% | 90.48% | 9.52% |

*Table 7.* Analysis of high-expert, high-amateur ("high-high") cases.

| Dataset | High-high ratio | Correct ratio | Expert top1 correct | GG-CRD top1 correct | Expert top1 correct retention | Avg. rank shift |
|---|---|---|---|---|---|---|
| Reefnot (cog.) | 2.25% | 97.22% | 96.30% | 97.22% | 100% | +0.92 |
| Reefnot (per.) | 5.63% | 70.24% | 66.94% | 70.24% | 98.34% | +3.30 |

## C. Experimental Analysis

### C.1. Quantitative Analysis of Amateur Scores and High-High Cases

To further validate the key assumptions behind GG-CRD, we provide two targeted quantitative analyses. First, we examine whether candidates assigned high scores by the amateur model are indeed more likely to correspond to hallucinated relations. Second, we analyze the challenging high-expert, high-amateur ("high-high") cases, where the contrastive formulation may potentially down-weight valid relations that are both visually grounded and consistent with prior bias.

**Amateur scores and hallucination likelihood.** we partition amateur-model scores into three equal-width bins over $[0, 1]$, and compute the proportions of hallucinated and correct relations within each bin. The results are reported in Table 6.

As shown in Table 6, the proportion of hallucinated relations increases monotonically with the amateur score on both datasets. On Reefnot_Cog, hallucinated relations account for 9.09%, 76.62%, and 94.03% in the low-score, medium-score, and high-score bins, respectively. A similar trend is observed on Reefnot_Per, where the corresponding proportions are 10.05%, 74.13%, and 90.48%. In particular, in the high-score bin, over 90% of the relations are hallucinated on both datasets.

These results provide direct empirical support for the assumption that the amateur-model score is strongly predictive of hallucination likelihood. Therefore, the amateur model does not merely act as an erroneous predictor, but instead captures bias-driven tendencies that are closely related to hallucinated relations.

**Stability of GG-CRD in high-high cases.** We further investigate a key scenario in which candidate relations receive high scores from both experts and novices. Although such "high–high" agreement does not necessarily indicate hallucination, it is still important to verify whether GG-CRD erroneously suppresses valid relations under this setting.

Several observations can be drawn from Table 7. First, high-high cases are relatively rare, accounting for only 2.25% and 5.63% of all candidates on Reefnot_Cog and Reefnot_Per, respectively. This indicates that their overall impact on the final prediction is limited. Second, correct relations dominate this subset, especially on Reefnot_Cog, where 97.22% of high-high cases are correct. Third, applying GG-CRD does not reduce top-1 correctness in this setting; instead, the top-1 accuracy slightly improves from 96.30% to 97.22% on Reefnot_Cog and from 66.94% to 70.24% on Reefnot_Per. Fourth, the retention rate of correct expert top-1 predictions remains extremely high, reaching 100% and 98.34% on the two datasets, respectively. Finally, although GG-CRD introduces some rank adjustment, the average rank shifts (+0.92 and +3.30) remain limited and do not lead to the loss of correct top-ranked predictions.

Overall, these results suggest that GG-CRD acts as a soft and stable bias-correction mechanism. Even in the high-high scenario, where valid relations may also receive strong amateur scores, GG-CRD largely preserves correct high-confidence predictions while still yielding net improvements in top-1 accuracy.

### C.2. Efficiency Analysis

We further evaluate the computational efficiency of our method on the VG-Zero dataset by comparing HSGG with OWSGG, a representative training-free scene graph generation method. The inference time per image is summarized in Table 8.

*Table 8.* Inference time comparison on the VG-Zero dataset.

| Method | Time (sec/image) |
|---|---|
| OWSGG | 59.8 |
| HSGG (Ours) | 61.3 |

OWSGG requires 59.8 seconds per image, while HSGG takes 61.3 seconds per image, showing that our method maintains comparable efficiency despite its richer structural modeling.

*Table 9.* Comparison of different methods on VG150 under the open-world setting. $*$ indicates results reproduced by us, as the original reported results were evaluated under the open-vocabulary setting.

| Method | Need Training? | R@20/50/100 | mR@20/50/100 |
|---|---|---|---|
| OwSGG | Training-free | - / 1.6 / 2.4 | 1.3 / 2.1 / 2.9 |
| R1SGG[*] | Supervised | 6.6 / 9.1 / 9.4 | 6.9 / 9.3 / 9.7 |
| SPADE[*] | Supervised | **9.8 / 12.9 / 15.5** | 7.9 / 9.6 / 12.5 |
| ACC[*] | Supervised | 7.8 / 10.3 / 13.1 | 6.9 / 8.9 / 10.6 |
| VL-IRM[*] | Supervised | 8.2 / 10.9 / 13.7 | 6.7 / 8.5 / 10.8 |
| HSGG (Ours) | Training-free | 6.4 / 9.3 / 11.9 | **8.5 / 10.2 / 13.1** |

## C.3. HSGG vs. Other SGG model: Performance Comparison

As shown in Table 9, under the open-world setting on the VG150 dataset, we compare our method with a series of approaches, including OWSGG (Dutta et al., 2025), R1SGG (Chen et al., 2025), SPADE (Hu et al., 2025), ACC (Li et al., 2026), and VL-IRM (Min et al., 2025). The experimental results show that, although our method is training-free, it achieves performance on the R@K metric that is comparable to those of supervised methods. Moreover, benefiting from the bias correction mechanism of the GG-CRD module, our method demonstrates strong capability in recognizing tail relationships, thereby achieving the best results on the mR@K metric.

*Table 10.* Comparison results on different inputs in Image caption.$*$ denotes our reproduced results.

| Input | $S_{\text{object}} \uparrow$ | $S_{\text{relation}} \uparrow$ | $S$-Cov $\uparrow$ |
|---|---|---|---|
| OwSGG | 74.36 | 2.71 | 54.20 |
| R1SGG[*] | 77.83 | 2.75 | 58.21 |
| SPADE[*] | 77.68 | 2.76 | 57.96 |
| ACC[*] | 75.22 | 2.75 | 55.95 |
| VL-IRM[*] | 76.70 | 2.73 | 54.89 |
| Image | 78.19 | 2.71 | 58.87 |
| HSG | **82.96** | **2.85** | **60.33** |

## C.4. Comparison of HSGG with Other SGG Models Input in Image Caption Applications

To further evaluate the utility of scene graphs generated by current SGG methods in downstream tasks, we compare different methods on image captioning, as summarized in Table 10. On the CompreCap dataset, our HSGG generated hierarchical scene graphs achieve the best performance, likely because their structured hierarchical representation captures image content more comprehensively.

## C.5. More Ablation Study

**Effect of Relation Filtering Threshold.** We analyze the sensitivity of the GA-RF module to the filtering threshold $\tau$ on a GPT-4o–annotated subset of the PSG dataset. As shown in Figure 5a, $\tau = 0.3$ provides a favorable trade-off, achieving a high recall of 76.56% while retaining a reasonable level of precision. Further increasing the threshold yields diminishing returns in recall, while the substantially increased noise level (precision below 10%) introduces unnecessary computational

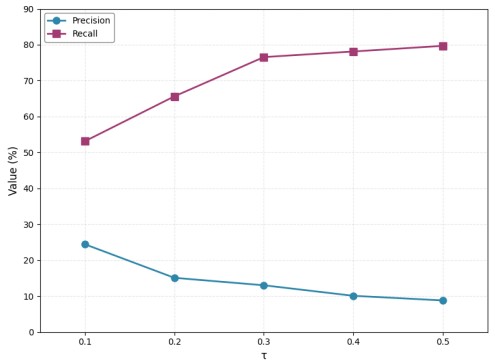

*(a)* Impact of the threshold $\tau$ on relation filtering performance.

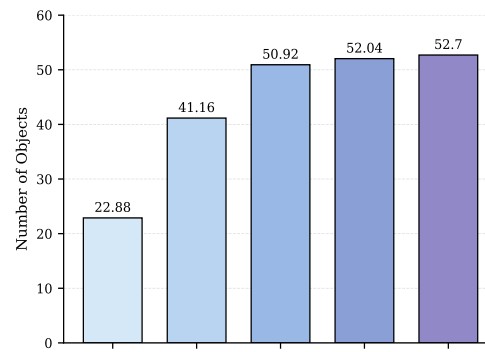

*(b)* Sensitivity analysis of hierarchy depth $\ell$ in HSGG.

*Figure 5.* Additional ablation studies. Left: effect of relation filtering threshold $\tau$. Right: sensitivity analysis of hierarchy depth $\ell$.

*Table 11.* Ablation on the contrast factor $\alpha$.

| $\alpha$ | Reefknot (Cog.) | Reefknot (Per.) |
|------|------|------|
| 0.75 | 89.3 | 62.08 |
| 1.00 | **90.0** | **63.07** |
| 1.25 | 88.7 | 62.11 |

*Table 12.* Ablation on the truncation threshold $\beta$.

| $\beta$ | Reefknot (Cog.) | Reefknot (Per.) |
|------|------|------|
| 0.0 | 88.2 | 61.87 |
| 0.1 | **90.0** | **63.07** |
| 0.2 | 89.7 | 62.87 |

overhead in subsequent relation reasoning. Based on this analysis, we set $\tau = 0.3$ for all experiments to balance relational coverage and robustness.

**Optimal Depth for Hierarchical Scene Graphs** As illustrated in Figure 5b, we examine the effect of hierarchy depth $\ell \in \{1, 2, 3, 4, 5\}$ on entity discovery. Increasing the depth from $\ell = 1$ to $\ell = 3$ substantially expands the scene graph, enabling the discovery of fine-grained sub-objects and long-tail entities. Further increasing the depth to $\ell = 4$ yields limited additional benefits, while introducing redundant and overlapping entity representations. From an empirical standpoint, $\ell = 3$ offers a favorable trade-off between semantic granularity and computational cost, and is therefore adopted as the default setting in HSGG.

**Effect of contrast factor.** Table 11 presents the impact of different contrastive coefficients on mitigation Relational Hallucinations on the Reefknot dataset. We observe that the overall performance is not highly sensitive to the choice of the hyperparameter $\alpha$.

**Effect of truncation threshold.** Table 12 presents the impact of different truncation threshold on mitigation Relational Hallucinations on the Reefknot dataset. We observe that setting $\beta = 0$, applying no truncation, leads to suboptimal performance. In contrast, excessive truncation also degrades performance. Based on this observation, we set $\beta = 0.1$ in all experiments.

## D. Visualization of HSG.

Scene graphs extracted from the VG dataset typically exhibit relatively flat and sparse semantic structures, capturing mostly surface-level object relations. In contrast, as illustrated in Figure 6, our method produces hierarchically organized scene graphs with substantially richer semantics. Beyond structural organization, we further leverage the generated HSGs to produce corresponding hierarchical descriptions, providing an additional semantic view of the scene. The three visualizations collectively demonstrate that, enabled by the HEP module, HSG supports fine-grained semantic decomposition across multiple abstraction levels, resulting in more structured and interpretable representations that better reflect human perceptual and cognitive organization. Such expressive multi-level encodings offer a promising foundation for downstream tasks requiring compositional and relational reasoning.

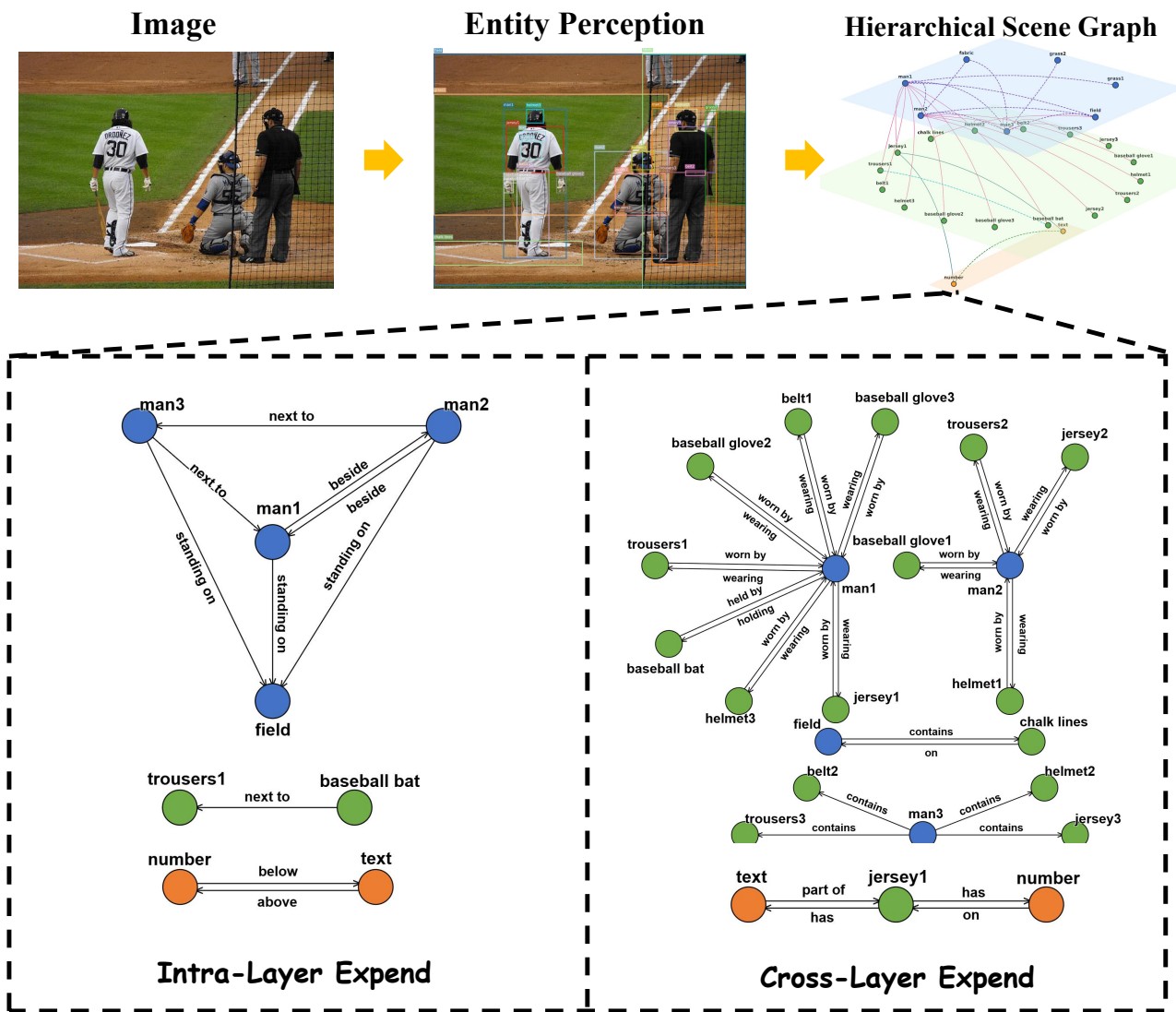

Description:The scene captures a moment during a professional baseball game, set against the backdrop of a meticulously maintained field that blends manicured green grass with dirt infield areas. The field is marked by thick white chalk lines and batter's box outlines, creating a clear boundary for the players. In the foreground, a black protective safety netting, known as backstop netting, frames the view of the action, adding a layer of depth to the scene.

At the heart of the scene, three key figures stand out. Magglio Ordóñez, the batter, stands at home plate, donning a white Detroit Tigers home jersey with "DROOÑEZ 30" emblazoned on the back. His attire includes white baseball pants with thin navy blue piping down the side, secured by a black leather athletic belt. He wears a black shiny batting helmet and holds a wooden baseball bat vertically, with the barrel resting on the ground. To his left, Russell Martin, the catcher, crouches in readiness, clad in a grey Los Angeles Dodgers away uniform complete with full protective gear. His outfit consists of grey baseball pants tucked into protective shin guards, a brown leather catcher's mitt on his left hand, and a blue protective catcher's helmet with a metal face cage. Behind the catcher, the home plate umpire stands alert, dressed in a black short-sleeved polo shirt and formal grey trousers, complemented by a black leather belt. The interplay between these characters, their positions, and their equipment paints a vivid picture of a tense moment in the game, where every detail contributes to the anticipation of the next play.

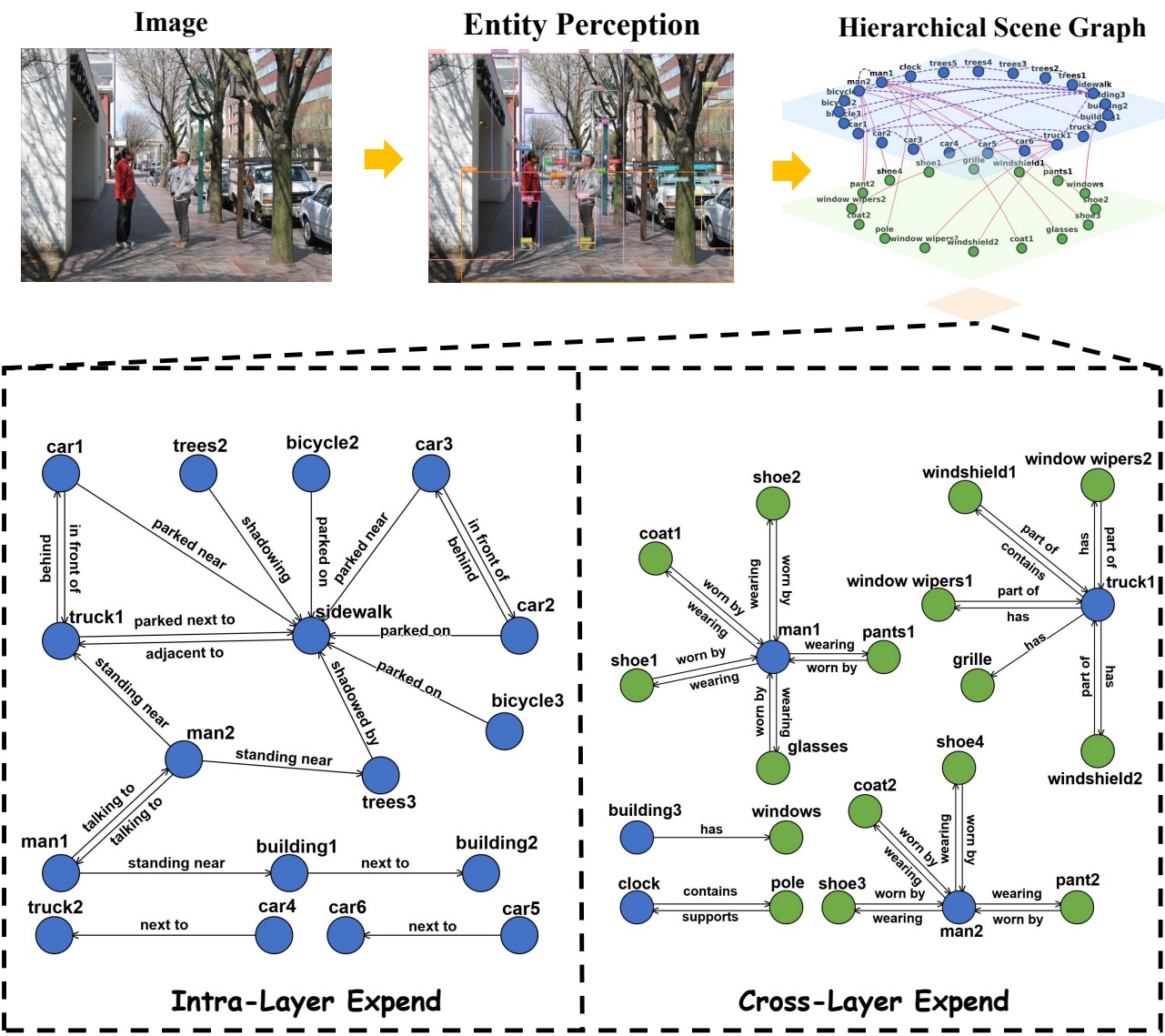

**Description**: The image captures a lively urban scene on a bustling city street. Two men stand engaged in conversation on a wide, dark reddish-brown brick sidewalk. One man wears a long-sleeved red shirt and black pants, while the other dons a grey zippered hoodie and dark grey casual pants. In the background, various vehicles add to the dynamic atmosphere, including a white car parked on the right side of the street, a silver or light-colored car further down the street, and a white pickup truck with a silver metal ladder rack or utility frame over its bed. Additionally, several bicycles are parked along the sidewalk, partially obscured by other objects. The scene is framed by buildings, trees, and a prominent green, double-sided vintage street clock standing on the sidewalk, supported by a tall green metal post. Details reveal a rich tapestry of relationships and attributes within the scene. The man in the red shirt stands near a white brick building with a black sign at the top, while the man in the grey hoodie gestures as he talks, standing near a thin deciduous tree and a larger tree in the foreground on the right, which frames the white car. The white pickup truck is parked next to the sidewalk, casting a shadow from the nearby trees. The green street clock, supported by a tall green metal post, adds a touch of nostalgia to the modern urban setting. The various vehicles, including cars and trucks, are parked or driving in different areas of the street, contributing to the sense of movement and activity. Overall, the scene is a vibrant snapshot of everyday life in the city, with each element playing a role in creating a cohesive and engaging narrative.

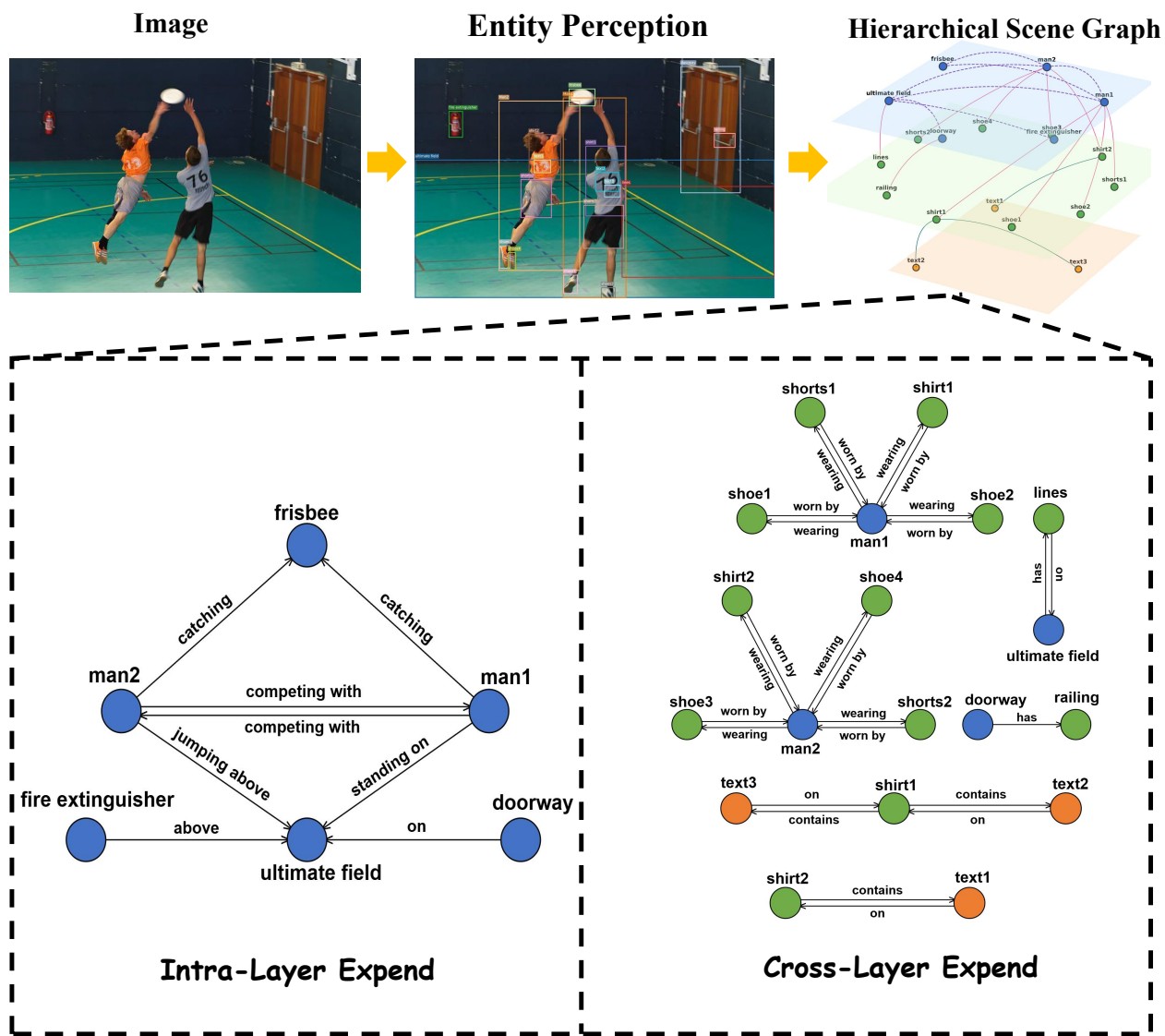

**Description**: In the midst of an intense indoor Ultimate Frisbee match, two athletes are captured in a dynamic moment on the green court marked with yellow and black lines. The athlete in the gray shirt, adorned with the number "76," is seen leaping with his arm extended, attempting to block the incoming white frisbee. Simultaneously, the competitor in the orange shirt, bearing the number "13," is also airborne, reaching out to catch the same frisbee. Both players are dressed in athletic attire suitable for the sport; the first wears black shorts while the second sports gray ones. Their footwear, specifically designed for indoor court traction, is visible as they both make their respective moves.

The court itself is bordered by a double wooden doorway featuring a metal horizontal door handle or crash bar. Mounted on the wall adjacent to the court is a red safety fire extinguisher, ready for any emergencies.

*Figure 6.* HSG Visualization

# E. Limitation and Future Work

Despite the strong effectiveness of HSGG in open-world hierarchical scene graph generation and hallucination mitigation, its performance naturally reflects the quality of the underlying vision-language and depth estimation components, as is common for training-free inference frameworks. While the current pipeline incurs computational costs comparable to existing scene graph generation methods, the structured hierarchical representations produced by HSGG offer opportunities for distilling its geometry-aware reasoning into more efficient models. Future work will focus on improving inference efficiency, for example through lightweight geometric adapters, and extending the proposed hierarchical paradigm to video scene graph generation for modeling complex spatiotemporal relationships.

