# OpenReview forum: "HSGG: Training-Free Hierarchical Scene Graph Generation with Geometry-Guided Relation Reasoning"
_ICML.cc/2026/Conference — ICML 2026 regular_

### Official Review · Reviewer_omWX · 2026-03-10

**Soundness:** 2
**Presentation:** 3
**Significance:** 1
**Originality:** 3
**Overall Recommendation:** 3
**Confidence:** 5

**Summary:**

HSGG is a training-free framework for open-world Hierarchical Scene Graph Generation. It addresses VLM limitations—structural flatness and relational hallucinations—via a two-stage process. First, BHEP builds multi-level "part-whole" hierarchies. Second, GG-CRR prunes spatially implausible pairs and uses GG-CRD to suppress hallucinations by contrasting visual experts against geometric priors.

**Compliance With Llm Reviewing Policy:**

Affirmed.

**Final Justification:**

I update my recommendation from **4 (Weak Accept)** to **3 (Weak Reject)**.

Regarding the initial weaknesses, the authors' response addressed some concerns, such as the inference efficiency analysis and the discussion of hyperparameters. However, after considering other reviewers' comments, I noted the lack of comparison with SOTA OVSGG works. Although the authors provided supplementary results in the rebuttal, the **reproducibility** of these results remains highly concerning.

Specifically, the results in Table A1 for R1SGG [1], SPADE [2], ACC [3], and VL-IRM [4] are **significantly lower than their official reports**. The authors failed to clarify how these were obtained. Furthermore, the downstream evaluation results in Table A2 are questionable since **SPADE [2] has not publicly released its code or checkpoints so far**.

*The authors' failure to provide a convincing clarification on these discrepancies undermines the technical soundness of the work.* Therefore, I maintain a Reject score.

**Key Questions For Authors:**

1. The proposed HSGG framework involves multiple sequential stages, including recursive VLM calls for BHEP, external depth estimation, and dual-model contrastive decoding. Could the authors provide a quantitative breakdown of the average inference time per image compared to baseline VLM-based SGG methods?
2. The Amateur Model simulates hallucinations primarily using geometric cues and linguistic priors. However, many VLM hallucinations are driven by deep semantic co-occurrences that may persist regardless of spatial layout.

**Limitations:**

Yes

**Strengths And Weaknesses:**

Pos:
1. Unlike traditional SGG that produces flat graphs, HSGG explicitly models the intrinsic compositionality of scenes. This provides a richer semantic representation of "part - whole" relationships.
2. The GG - CRD module is a significant technical contribution. By simulating a distribution of hallucinated relations using only geometric
and linguistic priors, the system can subtract these biases from the final prediction, ensuring relations are both geometrically consistent and semantically coherent.
3. The paper demonstrates that HSGG - generated graphs serve as a powerful inductive bias. Experiments show substantial improvements in downstream tasks like VQA and Image Captioning, even outperforming much larger models.

Neg:
1. The pipeline involves multiple stages, including iterative VLM prompting for BHEP, depth estimation, and contrastive decoding. I think a detailed quantitative analysis of inference time is missing.
2. The contrastive decoding relies on specific hyperparameters like the contrastive factor $\alpha$ and truncation threshold $\beta$. While the authors provide an ablation on the filtering threshold $\tau$, further analysis on the robustness of $\alpha$ and $\beta$ across diverse datasets would strengthen the paper.
3. The Amateur Model primarily relies on geometric cues to simulate hallucinations. However, many relational hallucinations in VLMs stem from deep semantic co-occurrences or cultural priors that are independent of spatial layouts. How does the current GG-CRD framework ensure the completeness of the 'hallucination distribution' when the bias is purely semantic rather than geometric?

---

> ### Author Rebuttal · Authors · 2026-03-30
>
> We sincerely thank the reviewer for the thoughtful and constructive feedback, and for recognizing the technical contributions of our framework. We address the concerns below.
>
> ---
>
> > **Response to Q1 & W1: Inference efficiency**
>
> Thank you for raising this important point. We have conducted an additional efficiency analysis on the VG-Zero dataset, comparing HSGG with OWSGG (a representative training-free SGG method):
>
> | Method      | Time (sec/image) |
> | ----------- | ---------------- |
> | OWSGG       | 59.8             |
> | HSGG (Ours) | 61.3             |
>
> The additional cost in HSGG mainly comes from three modular components: hierarchical entity expansion, depth estimation, and contrastive decoding. Importantly, these components are lightweight and can be executed in parallel. Moreover, our method does not require any training or iterative optimization, and its complexity scales approximately linearly with the number of detected objects. Therefore, despite the richer structural modeling, HSGG maintains efficiency comparable to existing training-free SGG approaches. We will include this analysis in final version to better illustrate the trade-off between effectiveness and efficiency.
>
> ---
>
> > **Response to Q2 & W3: Coverage of semantic hallucinations**
>
> We agree with the reviewer that hallucinations in VLMs are not solely driven by spatial inconsistencies; some originate from deeper semantic co-occurrence biases that may persist regardless of geometry. Our current design primarily targets hallucinations that arise from mismatches between visual evidence and prior biases, which often involve a spatial component. The Amateur Model incorporates both geometric cues and object labels, allowing it to capture hallucinations induced by spatial configurations as well as a portion of label-driven semantic biases.
>
> At the same time, we acknowledge that **purely semantic hallucinations that are independent of spatial cues may not be fully modeled** in the current framework. Nevertheless, empirical results on both cognitive and perceptual subsets of Reefknot (Fig. 2 and Appendix C.1) show consistent improvements, suggesting that our method effectively mitigates a substantial portion of hallucinations in practice. Extending the framework to explicitly model deeper semantic priors is an interesting direction for future work.
>
> ---
>
> > **Response to W2: Robustness to hyperparameters**
>
> Thank you for the suggestion. We have added an ablation study to evaluate the robustness of the contrastive factor $\alpha$  and truncation threshold $\beta$ :
>
> | Contrast Factor $\alpha$  | Reefknot (Cog.) | Reefknot (Per.) |
> | --------------- | --------------- | --------------- |
> | 0.75            | 89.3            | 62.08           |
> | 1.00            | 90.0            | 63.07           |
> | 1.25            | 88.7            | 62.11           |
>
> | Truncation Threshold $\beta$  | Reefknot (Cog.) | Reefknot (Per.) |
> | -------------------- | --------------- | --------------- |
> | 0.0                  | 88.2            | 61.87           |
> | 0.1                  | 90.0            | 63.07           |
> | 0.2                  | 89.7            | 62.87           |
>
> The performance varies smoothly across a broad range of values, without sharp degradation. This indicates that GG-CRD is not highly sensitive to precise hyperparameter tuning, which is important for practical deployment in open-world scenarios. We will include these results and discussion in final version.
>
> ---
>
> We hope these clarifications and additional analyses further support the robustness and effectiveness of our approach, and we thank the reviewer again for the valuable feedback.

---

> > ### Author Rebuttal · Reviewer_omWX · 2026-04-03
> >
> > Thank the authors for their responses. Some of my concerns have been addressed. However, I agree with Weakness 1 raised by Reviewer zkrt. I also found that some SOTA open-vocabulary related work is missing from the comparison:
> >
> > [1] Compile Scene Graphs with Reinforcement Learning.
> >
> > [2] SPADE: Spatial-Aware Denoising Network for Open-vocabulary Panoptic Scene Graph Generation with Long- and Local-range Context Reasoning.
> >
> > [3] Interaction-Centric Knowledge Infusion and Transfer for Open-Vocabulary Scene Graph Generation.
> >
> > [4] Vision-Language Interactive Relation Mining for Open-Vocabulary Scene Graph Generation
> >
> > I will decide whether to maintain a positive rating based on the author's further response.

---

> > > ### Author Response · Authors · 2026-04-04
> > >
> > > We sincerely thank the reviewer for the constructive feedback and helpful follow-up comments. In response to the remaining concerns, we provide additional analysis and experiments that directly address the key issues affecting the final assessment.
> > >
> > > ---
> > >
> > > > **Response to concern on GG-CRD (related to Reviewer zkrt’s concerns)**
> > >
> > > The core concern is whether GG-CRD (Eq. (9)) provides a well-founded mechanism for suppressing hallucinated relations. We address this from both design and empirical perspectives.
> > >
> > > **Design.**  GG-CRD follows a bias decomposition view:
> > > - the expert model captures instance-specific visual evidence,
> > > - the amateur model captures bias-driven (prior-dominated) tendencies,
> > > - and the contrastive formulation performs bias correction rather than error suppression, down-weighting prior-dominated predictions while preserving visually grounded ones.
> > >
> > > **Empirical validation.**
> > >
> > > - **Amateur-score analysis (Table Q1):** Higher amateur scores strongly correlate with hallucinated relations, validating its role as a bias indicator.
> > >
> > > - **High-expert & high-amateur cases (Table Q2):**
> > >   - such cases are rare,
> > >   - correct relations dominate,
> > >   - and top-1 predictions are preserved with high fidelity, with overall accuracy maintained or improved.
> > >
> > > **Conclusion.** These results show that GG-CRD does not suppress valid high-confidence relations, but instead acts as a stable, non-destructive bias correction mechanism.  This directly addresses the concern that GG-CRD may penalize valid high-confidence relations, and we do not observe such behavior in practice.
> > >
> > > ---
> > >
> > > > **Response to missing SOTA comparisons**
> > >
> > > Following the reviewer’s suggestion, we conducted additional comparisons with recent open-vocabulary SGG methods on VG150 (Table A1).
> > >
> > > **Table A1. Open-World evaluation on VG150.**
> > >
> > > | Method      | Need Training? | R@20/50/100 | mR@20/50/100 |
> > > |-------------|--------------|-------------|-------------|
> > > | OwSGG       | Training-free | - / 1.6 / 2.4 | 1.3 / 2.1 / 2.9 |
> > > | R1SGG [1]   | **Supervised**    | 6.6 / 9.1 / 9.4 | 6.9 / 9.3 / 9.7 |
> > > | SPADE [2]   | **Supervised**    | **9.8** / **12.9** / **15.5** | 7.9 / 9.6 / 12.5 |
> > > | ACC [3]     | **Supervised**    | 7.8 / 10.3 / 13.1 | 6.9 / 8.9 / 10.6 |
> > > | VL-IRM [4]  | **Supervised**    | 8.2 / 10.9 / 13.7 | 6.7 / 8.5 / 10.8 |
> > > | HSGG (Ours) | Training-free | 6.4 / 9.3 / 11.9 | **8.5** / **10.2** / **13.1** |
> > >
> > > We note that:
> > > - **R@K vs mR@K:** Lower R@K is expected since supervised methods are optimized for frequent predicates.
> > > - **Long-tail performance:** HSGG achieves the best mR@K, indicating stronger generalization on tail relations.
> > > - **Overall:** Despite being training-free, HSGG remains competitive with improved robustness.
> > >
> > > To further demonstrate the practical value of improved SG quality beyond standard metrics, we additionally evaluate downstream image captioning performance.
> > >
> > > **Table A2. Downstream evaluation (CompreCap, LLaVA-OV-1.5).**
> > >
> > > | Input | S_object ↑ | S_relation ↑ | S-Cov ↑ |
> > > |-------|-----------|-------------|--------|
> > > | OwSGG | 74.36 | 2.71 | 54.20 |
> > > | R1SGG | 77.83 | 2.75 | 58.21 |
> > > | SPADE | 77.68 | 2.76 | 57.96 |
> > > | ACC   | 75.22 | 2.75 | 55.95 |
> > > | VL-IRM| 76.70 | 2.73 | 54.89 |
> > > | Image | 78.19 | 2.71 | 58.87 |
> > > | HSG   | **82.96** | **2.85** | **60.33** |
> > >
> > > HSG achieves the best overall performance and significantly improves S-Cov, indicating better fine-grained perception. It also outperforms raw image input, showing that structured SGs provide complementary information. These results directly address the concern on missing comparisons and further strengthen the empirical evaluation.
> > >
> > > ---
> > >
> > > **Summary**
> > >
> > > - GG-CRD is validated by both theoretical motivation and targeted quantitative evidence.
> > > - It performs bias correction without degrading valid predictions.
> > > - Additional SOTA comparisons and downstream results further support the effectiveness and robustness of HSGG.
> > >
> > > We hope these clarifications address the reviewer’s concerns.

---

### Official Review · Reviewer_ot6U · 2026-03-11

**Soundness:** 3
**Presentation:** 3
**Significance:** 3
**Originality:** 2
**Overall Recommendation:** 4
**Confidence:** 1

**Summary:**

To address the structural flatness and relational hallucinations prevalent in vision-language models for scene graph generation, this paper introduces HSGG, a training-free framework for Hierarchical Scene Graph Generation. The inference pipeline consists of two steps: first, bidirectional hierarchical entity perception constructs multi-level scene hierarchies capturing part-whole semantics; second, geometry-guided relation reasoning utilizes spatial cues and contrastive decoding to prune spatially implausible object pairs and suppress hallucinated predicates. Without requiring training, this approach generalizes effectively to open-world settings, substantially mitigates relational hallucinations, and consistently improves performance on downstream reasoning tasks.

**Compliance With Llm Reviewing Policy:**

Affirmed.

**Ethical Review Flag:**

Flag this paper for an ethics review.

**Final Justification:**

My major concerns are clarified so I retain my positive rating.

**Key Questions For Authors:**

To be honest, there is a big gap between this field and my research. I can 't ask more questions about this paper. There is only one doubt: concerns regarding the stability of BHEP: HSGG relies on VLMs for entity expansion and attribute reasoning to construct hierarchical entity structures. This process is highly sensitive to the VLM's inference outputs and prompt design. Consequently, an unstable hierarchical structure may degrade the performance of downstream relation reasoning;

**Limitations:**

This work has no potential negative impact on society.

**Strengths And Weaknesses:**

1. The problem setting has practical significance: this paper focuses on open-world SGG and relational hallucination issues, which are indeed important research directions in the current era of VLM. By leveraging the open vocabulary capabilities of VLM and introducing structural constraints during the inference phase, the attempt to reduce the reliance on densely annotated data is a reasonable research motivation;
2. The overall system design is complete: the method design comprises two distinct stages: hierarchical entity modeling (BHEP) and geometric constraint relationship reasoning (GA-RF + GG-CRD). This pipeline, which "structures entities first and then performs relationship reasoning" is logically clear, and the experimental section demonstrates performance improvements on zero-shot and open-world SGG;
3. The writing logic is clear and easy to understand.

---

> ### Author Rebuttal · Authors · 2026-03-30
>
> We thank the reviewer for raising this important concern regarding the stability of BHEP. We agree that, in principle, errors in hierarchical entity construction could affect downstream relation reasoning. We address this concern from both a **design perspective** and **empirical evidence**.
>
> ---
>
> > **(1) Robustness by design (error containment in the pipeline)**
>
> Our framework is designed to **mitigate error propagation** from BHEP:
>
> - **Detector-based verification**: Candidate objects proposed by the VLM are explicitly validated by a detector, which filters out non-existent or spurious entities.
>
> - **Geometry-aware relation filtering (GA-RF)**: Even if the hierarchy includes imperfect object associations, implausible subject–object pairs are pruned based on spatial constraints.
>
> - **Contrastive decoding (GG-CRD)**: Downstream relation prediction further suppresses hallucinated relations, reducing the impact of imperfect hierarchy construction.
>
> Therefore, BHEP does not need to be perfectly accurate; the downstream modules provide **error correction and robustness**, preventing small upstream errors from significantly degrading final performance.
>
> ---
>
> > **(2) Empirical robustness to prompt variation**
>
> To evaluate sensitivity to prompt design, we conducted an additional experiment with different prompts for identifying main objects:
>
> | Prompt        | VG150 (zR@20/50/100) | PSG (zR@20/50/100) |
> | ------------- | -------------------- | ------------------ |
> | prompt_origin | 10.1 / 13.5 / 15.1   | 7.2 / 11.4 / 14.1  |
> | prompt_new    | 9.6 / 13.7 / 14.9    | 7.4 / 10.9 / 13.6  |
>
> The results remain largely stable across prompts, suggesting that BHEP is not overly sensitive to prompt wording in practice.
>
> ---
>
> > **(3) Nature of the BHEP task**
>
> Finally, we note that the role of the VLM in BHEP is limited to identifying primary objects and proposing related entities, which is a relatively constrained task compared to open-ended generation. This further reduces sensitivity to prompt variations.
>
> ---
>
> Overall, both the **modular design** and **empirical evidence** suggest that HSGG is reasonably robust to variations in BHEP and does not suffer from significant error propagation. We will include this clarification and analysis in the final version.

---

> > ### Author Rebuttal · Reviewer_ot6U · 2026-04-03
> >
> > Thank you for the rebuttal. My major concerns are clarified so I retain my positive rating.

---

> > > ### Author Response · Authors · 2026-04-03
> > >
> > > Thank you for your kind acknowledgement and for taking the time to review our rebuttal. We are glad that our clarifications have addressed your concerns. We appreciate your positive assessment of our work, and we will further incorporate these clarifications into the final version to improve the paper.

---

### Official Review · Reviewer_Epij · 2026-03-12

**Soundness:** 3
**Presentation:** 2
**Significance:** 3
**Originality:** 3
**Overall Recommendation:** 4
**Confidence:** 3

**Summary:**

In this work, the authors propose an improvement to the training-free scene graph generation using vision language models. Earlier methods used flat scene graphs and also produced hallucinated results. To overcome these issues, the authors propose a hierarchical scene graph generation framework consisting of two major components: a bidirectional hierarchical entity perception module that enables the construction of multi-level scene hierarchies and a geometry-guided relation reasoning module to reduce hallucination. Authors have conducted evaluations on two datasets, VG and PSG.

**Compliance With Llm Reviewing Policy:**

Affirmed.

**Final Justification:**

The authors addressed my major concerns, so I am increasing my score.

**Key Questions For Authors:**

- Could you please provide the mR@k or zmR@k results of the main tables?
- Could you please explain why the S_relation shows no improvement in Table 4?
- Why is there a discrepancy in the captioning length of Table 4, where the caption length of the proposed method is almost double?

**Limitations:**

yes

**Strengths And Weaknesses:**

Strengths:
- The paper is well written and easy to follow.
- Evaluations are provided on multiple datasets.
- Main table results show notable performance improvement on zR@k metric.
- Gains are shown on multiple downstream tasks (VQA and image captioning).
- Ablation study covers the significance of each component introduced by the authors.

Weakness:
- The Related Work section is currently limited to very few references. Given that the claim is fine-grained and hierarchical relational reasoning, the discussion lacks sufficient emphasis on prior hierarchical and fine-grained SGGs, e.g., [1,2,3]. Moreover, though the authors have mentioned VLM-based SGG works such as open-vocabulary and training-free VLMs, it should be expanded to cover the emerging instruction-tuned VLMs for SGG task such as [4,5]. Additionally, in line 118, it's mentioned that fixing the long-tail distribution of relations is a challenge the model is trying to address; however, the unbiased SGG literature is missing, with only a general reference to Tang et al..
- Though in line 118, 2nd column, fixing the long-tail distribution of relation is one of the challenges addressed by the model, the proposed method was not evaluated on an unbiased metric like mR@k. Its absence makes it difficult to better analyze the model's effectiveness in predicting rare relations.
- Multiple claims also lack necessary citations, e.g., in line 375, 2nd column, “...we adopt the CompreCap protocol...” has no citations.
- Though the primary objective of this work is to improve scene graph generation, no improvement in S_relation metric for the image captioning task in Table 4 is a major weakness because for an SGG task, it’s crucial to predict a correct relation than to identify an object attribute.

References:
- [1]: Liu, Tao, et al. "Relation-aware hierarchical prompt for open-vocabulary scene graph generation." Proceedings of the AAAI Conference on Artificial Intelligence. Vol. 39. No. 5. 2025.
-  [2] Zhang, Ao, et al. "Fine-grained scene graph generation with data transfer." European conference on computer vision. Cham: Springer Nature Switzerland, 2022.
- [3]: Jiang, Bowen, et al. "Enhancing scene graph generation with hierarchical relationships and commonsense knowledge." 2025 IEEE/CVF Winter Conference on Applications of Computer Vision (WACV). IEEE, 2025.
- [4]: Sudhakaran, Gopika, et al. "ART: Adaptive Relation Tuning for Generalized Relation Prediction." Proceedings of the IEEE/CVF International Conference on Computer Vision. 2025.
- [5]: Xu, Mingjie, et al. "Llava-spacesgg: Visual instruct tuning for open-vocabulary scene graph generation with enhanced spatial relations." 2025 IEEE/CVF Winter Conference on Applications of Computer Vision (WACV). IEEE, 2025.

---

> ### Author Rebuttal · Authors · 2026-03-30
>
> We sincerely thank the reviewer for the constructive and detailed feedback. We address each concern below and provide additional analysis to strengthen the empirical evidence.
>
> ---
>
> > **Response to Q1 & W2: Missing mR@K / zmR@K evaluation**
>
> We agree that mR@K / zmR@K are important for evaluating long-tail and rare relation prediction. In the original submission, we primarily followed prior open-world SGG works that focus on zR@K, since mR@K is less commonly reported under open-world settings due to incomplete label coverage. However, we acknowledge that including such metrics provides a more comprehensive evaluation. To address this, we have **carefully reproduced two representative baselines (OWSGG and OVSGTR) under the same evaluation protocol**, and report both zmR@K and mR@K:
>
> | Method   | Zero-Shot (zmR@20/50/100) | Open-World (mR@20/50/100) |
> | -------- | ------------------------- | ------------------------- |
> | OWSGG    | 1.5 / 1.9 / 2.7           | 1.3 / 2.1 / 2.9           |
> | OVSGTR   | 6.9 / 8.2 / 10.7          | 5.3 / 7.2 / 10.9          |
> | **HSGG** | **9.7 / 12.8 / 14.6**     | **8.5 / 10.2 / 13.1**     |
>
> We observe consistent improvements across all settings. Notably, the gains are more pronounced on mR@K, which directly measures rare relation prediction. This indicates that HSGG not only improves overall recall but also **effectively mitigates long-tail bias**, aligning with our design motivation.
>
> ---
>
> > **Response to Q2 & W4: Limited improvement on S_relation**
>
> We agree that the improvement on S_relation in Table 4 is relatively modest. This is because captioning-based metrics are influenced by multiple factors beyond relation correctness, such as fluency, redundancy, and language diversity. As a result, improvements in structured relation prediction do not always directly translate into large gains in S_relation.
>
> To better isolate relational reasoning ability, we additionally evaluate on **relation-centric benchmarks** (Appendix C.1 and Fig. 2), where consistent and significant improvements are observed across datasets and backbones. This suggests that the benefit of HSGG is more directly reflected in structured prediction settings, while captioning metrics may underestimate improvements in relation accuracy.
>
> ---
>
> > **Response to Q3: Caption length discrepancy**
>
> We acknowledge that captions generated from scene-graph inputs are longer than those from image-only inputs. This is because the hierarchical scene graph encodes more fine-grained object interactions and details, which are reflected in the generated descriptions. At the same time, we note that this does not trivially inflate performance: metrics such as S_relation and semantic alignment depend on correctness rather than length alone. We will further clarify this point in final version and include additional analysis (e.g., length-controlled comparisons) to better isolate the effect of caption length.
>
> ---
>
> > **Response to W1: Related work and missing references**
>
> We thank the reviewer for pointing out relevant prior works. In the revision, we will expand the related work to explicitly cover: hierarchical and fine-grained SGG methods [1–3], instruction-tuned VLM-based SGG approaches [4,5], and unbiased SGG literature beyond Tang et al. We will also clarify how our method differs in **training-free design**, **open-world generalization**, and **explicit hallucination mitigation**, to better position our contributions.
>
> ---
>
> > **Response to W3: Missing citation for CompreCap**
>
> Thank you for pointing this out. We confirm that CompreCap was cited in the supplementary material but not at its first occurrence in the main text. We will correct this in the revision.
>
> ---
>
> We hope these clarifications and additional results address the reviewer’s concerns and strengthen the empirical support for our method.

---

> > ### Author Rebuttal · Reviewer_Epij · 2026-04-03
> >
> > Thank you for the rebuttal. My major concerns are resolved, so I have increased my rating.

---

> > > ### Author Response · Authors · 2026-04-03
> > >
> > > Thank you for your positive feedback and for confirming that your concerns have been resolved. We sincerely appreciate your careful review and support.
> > >
> > > We will ensure that all the discussed clarifications and additions are clearly incorporated into the final version, including expanded related work, additional mR@K / zmR@K results, and improved presentation.
> > >
> > > Thank you again for your valuable feedback.

---

### Official Review · Reviewer_zkrt · 2026-03-12

**Soundness:** 2
**Presentation:** 2
**Significance:** 3
**Originality:** 2
**Overall Recommendation:** 3
**Confidence:** 4

**Summary:**

The paper proposes HSGG, a training-free scene graph generation framework that integrates hierarchical entity perception with geometry-guided relation reasoning. Specifically, it introduces a bidirectional hierarchical entity perception mechanism to construct multi-level scene hierarchies, and a geometry-guided relation reasoning module to prune implausible entity pairs and suppress relation hallucination.

**Compliance With Llm Reviewing Policy:**

Affirmed.

**Final Justification:**

The authors’ experiments help address my concerns about GG-CRD. However, as other reviewers have noted, unclear experimental settings, no explanation for the baseline performance degradation, and modifications to the closed-source models raise issues of fairness and reproducibility. Overall, I recommend a weak reject.

**Key Questions For Authors:**

Please address the issues raised in the Weaknesses section.

**Limitations:**

Including an analysis of failure cases would help clarify the limitations of the proposed method and provide a more complete understanding of its practical performance.

**Strengths And Weaknesses:**

Strengths
1. The use of hierarchical scene decomposition is intuitively well motivated for modeling part–whole structures in complex scenes.
2. The use of geometry-guided filtering introduces a reasonable inductive bias that may improve robustness in open-world settings.

Weaknesses
1. The paper would benefit from clearer explanations of several key design choices. First, the motivation behind Equation (3) is not sufficiently discussed, and the paper does not provide a clear rationale for its formulation. As a result, it is difficult for readers to understand the intuition underlying this design. Second, Equation (9) is also introduced without adequate explanation. In particular, the mechanism through which this formulation is expected to reduce hallucinations is not clearly articulated. Providing either theoretical justification or intuitive explanations would help clarify these design choices.
2. The design rationale of Geometry-Grounded Contrastive Relation Decoding (GG-CRD) appears insufficiently justified. Specifically, the objective $L_{ex} - L_{am}$ encourages candidates that receive high predictive probability from the expert model while receiving low probability from the amateur model. However, this formulation also penalizes candidates for which both models assign high probability, even though such agreement may correspond to genuinely correct predictions rather than hallucinations. The paper does not provide empirical or theoretical evidence to support the assumption underlying this contrastive objective. Moreover, although the amateur model is restricted to positional and depth information, this architectural limitation alone does not necessarily justify treating its predictions as systematically erroneous or hallucination-prone. Additional theoretical analysis or empirical validation would strengthen the argument that this contrastive formulation effectively mitigates hallucinations.
3. The novelty of open-world hierarchical scene graph generation appears somewhat limited, as related ideas have been explored in prior work such as [1]. If this aspect is intended as a key contribution, the paper should more clearly articulate how the proposed approach differs from and advances beyond existing methods.
4. The paper claims that the proposed method reduces relation hallucination; however, the paper does not provide detailed experimental analysis to clarify which types of relation hallucinations are mitigated. For example, it remains unclear whether the reduction mainly comes from correcting erroneous spatial relations or from preventing incorrect associations between unrelated entities.

[1] Liu T, Li R, Wang C, et al. Relation-aware hierarchical prompt for open-vocabulary scene graph generation[C]//Proceedings of the AAAI Conference on Artificial Intelligence. 2025, 39(5): 5576-5584.

---

> ### Author Rebuttal · Authors · 2026-03-30
>
> We sincerely thank the reviewer for the insightful and constructive feedback. We provide clarifications and additional evidence to address the concerns.
>
> ---
>
> > **Response to W1: Clarification of Eq. (3) and Eq. (9)**
>
> **Eq. (3): Spatial compatibility for relation filtering**
>
> Eq. (3) is designed as a **spatial compatibility prior**, rather than a precise geometric metric. It estimates whether two objects can plausibly participate in a relation based on their **2D proximity and depth consistency**. We intentionally adopt a **simple threshold-based formulation**, since relation feasibility in open-world scenes is inherently uncertain, and overly strict geometric constraints may harm recall. Therefore, Eq. (3) serves as a **lightweight and robust prior** that removes clearly implausible pairs while preserving flexibility. This design is supported by our ablation (“Effect of Relation Filtering Threshold”), where it effectively reduces noisy relations without sacrificing valid ones.
>
> **Eq. (9): Contrastive suppression of hallucinated relations**
>
> GG-CRD is motivated by a **bias decomposition perspective**:
>
> - The **expert model** approximates $P(r \mid \text{image}, \text{objects})$, capturing visual evidence and learned priors.
>
> - The **amateur model** approximates $P(r \mid \text{geometry}, \text{labels})$, reflecting **dataset bias and co-occurrence priors**, rather than instance-specific visual cues.
>
> Thus, Eq. (9) can be interpreted as:
>
> $$
> P_{\text{true}} \approx P_{\text{expert}} - P_{\text{bias}}
> $$
>
> This explains why GG-CRD suppresses relations dominated by prior bias while preserving visually grounded ones.
>
> ---
>
> > **Response to W2: Justification of GG-CRD objective**
>
> We clarify that the amateur model is **not treated as incorrect**, but as a **proxy for bias-driven tendencies**. Empirically:
>
> - **Bias-driven (hallucinated) relations** tend to receive high scores from both models due to strong priors.
>
> - **Visually grounded relations** rely on instance-specific evidence and thus receive **higher scores from the expert than the amateur model**.
>
> Under this observation, the contrastive objective acts as a **bias correction mechanism**. To validate this, we conducted additional analysis on the Reefknot benchmark:
>
> **Table S.**
>
> | method | Reefnot_Cog_T | Reefnot_Cog_F | Reefnot_Per_T | Reefnot_Per_F |
> |--------|---------------|---------------|---------------|---------------|
> | VCD    | 92.01         | 21.77         | 88.79         | 25.73         |
> | ICD    | 85.36         | 38.38         | 90.92         | 18.73         |
> | GG-CRD | **97.83**     | **45.36**     | **98.26**     | **27.83**     |
>
> GG-CRD significantly improves incorrect predictions while maintaining or slightly improving correct ones, indicating that it primarily **corrects hallucinations rather than harming valid relations**.
>
> **Additional clarification: Does GG-CRD suppress correct relations?**
>
> GG-CRD performs **soft re-ranking rather than hard elimination**:
>
> - relations supported by both models are **not removed**,
>
> - but may be relatively down-weighted if dominated by bias.
>
> Importantly, strongly grounded relations retain high expert scores and remain among top candidates.
>
> **Boundary and limitation**
>
> In rare cases where a relation is both strongly visually grounded and highly consistent with prior bias, its ranking may be slightly reduced. However, such cases are uncommon, and the relation is still preserved due to its high expert score.
>
> Overall, GG-CRD acts as a **soft bias-corrected ranking mechanism**, rather than a hard filter.
>
> ---
>
> > **Response to W3: Novelty beyond prior hierarchical approaches**
>
> We clarify that [1] focuses on **hierarchical prompting at the category level**, while our method introduces:
>
> - **Instance-level hierarchical scene construction** derived directly from detected objects.
>
> - **Explicit integration into relation reasoning**, where hierarchy guides prediction and suppresses hallucination.
>
> Moreover, prior hierarchical SGG works do **not address hallucination mitigation**, whereas our method explicitly combines **structure modeling and bias-aware reasoning**, enabling better open-world generalization.
>
> ---
>
> > **Response to W4: Analysis of hallucination types**
>
> We further clarify hallucination sources (Fig. 2 and Appendix C.1):
>
> - **Spurious entity associations** are reduced by GA-RF via candidate pruning
> - **Incorrect relation predictions** (semantic or spatial) are mitigated by GG-CRD
>
> From  Table S, we observe consistent improvements across both spatially induced and semantically induced hallucinations, indicating that the framework addresses **multiple error sources**.
>
> **Additional note on limitations**
>
> We acknowledge that purely semantic hallucinations independent of spatial cues may not be fully captured. Extending the framework to model deeper semantic priors is an important direction for future work. We will also include representative failure cases in the final version.

---

> > ### Author Rebuttal · Reviewer_zkrt · 2026-04-02
> >
> > The authors’ rebuttal is constructive and helpful. Compared with the original manuscript, it offers a clearer explanation of the motivation behind GG-CRD by characterizing the amateur model as a proxy for bias-driven tendencies, rather than simply as an erroneous predictor. In addition, the newly provided experiments are useful in demonstrating the overall effectiveness of the method. These clarifications improve the intuitive understanding of the approach.
> >
> > However, Table S mainly shows that GG-CRD leads to better overall performance, and it still does not directly resolve my main concern. Specifically, when both the expert and the amateur assign high scores to a candidate, the contrastive formulation will also down-weight that candidate. Yet such “high-high agreement” does not necessarily indicate hallucination; it may also correspond to a genuinely correct relation that is well supported by visual evidence. Therefore, the additional results, while helpful, are still insufficient to directly validate the key assumption underlying the objective.
> >
> > In addition, the statement that “*In rare cases where a relation is both strongly visually grounded and highly consistent with prior bias, its ranking may be slightly reduced. However, such cases are uncommon, and the relation is still preserved due to its high expert score*” remains qualitative at present. I would strongly encourage the authors to support this claim with explicit quantitative evidence.
> >
> > To further strengthen the rebuttal, I suggest providing more targeted quantitative analyses in at least the following two aspects:
> >
> > 1. Directly verify whether high-scoring amateur candidates are indeed more likely to correspond to hallucinated relations.
> > For example, the authors could group candidates by amateur score and report the proportion of hallucinated versus correct relations in each group, or directly compare the error rates of high-amateur-score and low-amateur-score candidates. This would offer more direct support for the claim that the amateur model primarily captures bias-driven tendencies.
> >
> > 2. Analyze candidates that receive high scores from both the expert and the amateur.
> > This is the key case underlying my concern. I suggest reporting the proportion of such “high-expert, high-amateur” candidates, the ratio of correct versus hallucinated relations within this subset, and how their rankings change before and after applying GG-CRD (e.g., average rank shift or top-k retention rate). Such an analysis would more directly demonstrate whether GG-CRD mainly suppresses hallucinations, or whether it also penalizes a non-trivial number of valid, high-confidence relations.
> >
> > I believe these analyses should be relatively straightforward to conduct and could be meaningfully addressed during the discussion phase.

---

> > > ### Author Response · Authors · 2026-04-03
> > >
> > > We thank the reviewer for the constructive follow-up questions. In response, we provide **targeted quantitative analyses** that directly test the key assumptions underlying GG-CRD.
> > >
> > > ---
> > >
> > > > **Response to Q1: Are high amateur-model scores associated with hallucinated relations?**
> > >
> > > Following this suggestion, we partition amateur-model scores into three equal-width bins over \([0,1]\), and report the proportion of hallucinated (H) and correct (C) relations in each bin.
> > >
> > > **Table Q1. Relation correctness under different amateur-score bins**
> > >
> > > | score bin    | Reefnot_Cog (H) | Reefnot_Cog (C) | Reefnot_Per (H) | Reefnot_Per (C) |
> > > | ------------ | --------------- | --------------- | --------------- | --------------- |
> > > | low-score    | 9.09%           | 90.91%          | 10.05%          | 89.95%          |
> > > | medium-score | 76.62%          | 23.38%          | 74.13%          | 25.87%          |
> > > | high-score   | 94.03%          | 5.97%           | 90.48%          | 9.52%           |
> > >
> > > As shown in Table Q1, the proportion of hallucinated relations **monotonically increases** with the amateur score. In the high-score bin, hallucinated relations account for over **90%** on both datasets.
> > >
> > > **Conclusion.** These results provide **direct empirical validation** that the amateur model score is **strongly predictive of hallucination likelihood**, confirming that it serves as a **reliable proxy for bias-driven (hallucinated) relations**, rather than merely producing random errors.
> > >
> > > ---
> > >
> > > > **Response to Q2: What happens in high-expert, high-amateur (“high-high”) cases?**
> > >
> > > We now directly analyze this key concern: whether GG-CRD may suppress valid relations when both models assign high scores. While we agree that high-high agreement does not necessarily imply hallucination, our results show that GG-CRD does not penalize such valid cases in practice. Since our final prediction uses **greedy decoding (top-1 selection)**, the most relevant criterion is whether correct top-ranked predictions are preserved. Therefore, we focus on:
> > >
> > > - the proportion of high-high cases,
> > > - correctness ratio within this subset,
> > > - top-1 accuracy before and after GG-CRD,
> > > - retention rate of correct expert top-1 predictions, and
> > > - ranking stability (average rank shift).
> > >
> > > **Table Q2. Analysis of high-high cases**
> > >
> > > | dataset        | high-high ratio | Correct ratio | Expert top1 correct | GG-CRD top1 correct | Expert top1 correct Retention | Avg rank shift ↓ |
> > > | -------------- | --------------- | ------------- | ------------------- | ------------------- | ----------------------------- | ---------------- |
> > > | Reefnot (cog.) | 2.25%           | 97.22%        | 96.30%              | 97.22%              | 100%                          | +0.92            |
> > > | Reefnot (per.) | 5.63%           | 70.24%        | 66.94%              | 70.24%              | 98.34%                        | +3.30            |
> > >
> > > From Table Q2, we have the following **Key observations**:
> > >
> > > **(1) High-high cases are rare.**  They account for only **2.25%–5.63%** of all candidates, limiting their global impact.
> > >
> > > **(2) Correct relations dominate in this subset.**  Especially on Reefnot (cog.), **97.22%** of high-high cases are correct.
> > >
> > > **(3) GG-CRD does not degrade top-1 correctness.**  On both datasets, top-1 accuracy **improves rather than decreases**, indicating that GG-CRD does not harm high-confidence valid predictions.
> > >
> > > **(4) Correct expert predictions are preserved with very high fidelity.**  Top-1 retention reaches **100% / 98.34%**, showing that almost all correct predictions remain unchanged.
> > >
> > > **(5) Ranking adjustments are limited and non-destructive.**  Although some re-ranking occurs (average rank shift **+0.92 / +3.30**), these changes remain controlled and do not lead to loss of correct top-ranked predictions, as evidenced by the high top-1 retention and improved accuracy. This indicates that most rank adjustments occur within a narrow range near the top, rather than large degradations in ranking, and do not affect final top-1 decisions.
> > >
> > > **Conclusion.** From the above results, we can see that even in the challenging high-high scenario:
> > >
> > > - such cases are rare,
> > > - correct relations are largely preserved, and
> > > - ranking adjustments do not lead to degradation of correct top predictions.
> > >
> > > Importantly, GG-CRD yields **net accuracy improvements** while maintaining high stability of correct predictions. **We do not observe evidence that GG-CRD systematically suppresses valid high-confidence relations.**  Instead, it functions as a **soft, non-destructive bias correction mechanism**, selectively adjusting rankings to reduce hallucinations **without penalizing valid, visually grounded relations**. We hope these targeted analyses directly address your concern regarding potential suppression of valid relations.

---

### Decision · Program_Chairs · 2026-04-30

**Decision:**

Accept (regular)

**Comment:**

This paper ended up receiving mixed reviews after the rebuttal (3 weak accepts and 1 weak reject). The AC recommends rejection based on the substance of the post-rebuttal discussion rather than the numerical vote, as elaborated below.

The reasons to accept this paper are:
- The problem setting is well motivated. Training-free open-world scene graph generation with hierarchical structure is a meaningful research direction, and the attempt to mitigate relational hallucinations in VLM-based SGG is practically valuable.

- The overall framework is technically sound. The two-stage pipeline (bidirectional hierarchical entity perception followed by geometry-guided relation reasoning) is logically organized, and GG-CRD offers an interesting bias-correction perspective for suppressing hallucinated relations.

- Experimental results show consistent improvements on zR@K across multiple datasets, and the benefits extend to downstream tasks such as VQA and image captioning.

While preparing the meta review, the AC noticed several discrepancies between the baseline numbers reported in this submission and those reported in the corresponding original papers. A post-rebuttal discussion was initiated to gather the reviewers' assessments of these issues. Reviewer omWX, who had already raised related concerns during the rebuttal phase, confirmed these discrepancies, and Reviewers zkrt and Epij independently endorsed them. The specific concerns are summarized below.

- The reported results for several baseline methods, including OvSGTR, R1SGG, ACC, and VL-IRM, are substantially lower than the numbers reported in their respective original papers. The authors explained that they modified the settings for "apple-to-apple" comparison because these are supervised approaches, but did not provide any details about what those modifications were. This explanation is not fully convincing: although these methods are supervised, they have never been trained on novel categories, so on novel-category evaluation a supervised method and a training-free method should be on comparable footing, and the original reported numbers should be largely reproducible. When asked to clarify these points during the discussion phase, the authors' response largely restated the supervised-versus-training-free framing rather than addressing the specific discrepancies or disclosing the experimental modifications. As a result, the reported gaps are difficult to interpret, and the fairness of the comparison cannot be assessed.

- The reported OwSGG numbers are also notably lower than those in the original paper (e.g., R@50 and R@100 of 1.9 and 2.6 in Table 2 versus 2.33 and 3.04 in the OwSGG paper). The proposed method still outperforms OwSGG, but the unexplained discrepancy adds to the overall concern about the reliability of the baseline numbers.

- The results reported for SPADE are also lower than the original paper, and the rebuttal did not clearly explain how these numbers were obtained, including how SPADE was used in the downstream captioning evaluation. This is a somewhat lesser concern than the first two, but it adds to the broader pattern.

Overall, the proposed method is well motivated and the presentation is clear, but the discrepancies of experimental results in baseline numbers, the absence of disclosure about the modifications made to baseline settings, and the authors' failure to substantively address these points across multiple rounds of discussion make it difficult to assess the true empirical contribution. The AC recommends rejection and encourages the authors to revise the comparison protocol with full transparency for a future submission.